# Heat health risk assessment in Philippine cities using remotely sensed data and social-ecological indicators

Ronald C. Estoque [1✉], Makoto Ooba[1], Xerxes T. Seposo[2], Takuya Togawa[1], Yasuaki Hijioka[1], Kiyoshi Takahashi [1] & Shogo Nakamura[1]

More than half of the world's population currently live in urban areas and are particularly at risk from the combined effects of the urban heat island phenomenon and heat increases due to climate change. Here, by using remotely sensed surface temperature data and social-ecological indicators, focusing on the hot dry season, and applying the risk framework of the Intergovernmental Panel on Climate Change, we assessed the current heat health risk in 139 Philippine cities, which account for about 40% of the country's total population. The cities at high or very high risk are found in Metro Manila, where levels of heat hazard and exposure are high. The most vulnerable cities are, however, found mainly outside the national capital region, where sensitivity is higher and capacity to cope and adapt is lower. Cities with high levels of heat vulnerability and exposure must be prioritized for adaptation. Our results will contribute to risk profiling in the Philippines and to the understanding of city-level heat health risks in developing regions of the Asia-Pacific.

--------

[1] National Institute for Environmental Studies, 16-2 Onogawa, Tsukuba City, Ibaraki 305-8506, Japan. [2] Nagasaki University, 1-12-4 Sakamoto, Nagasaki City, Nagasaki 852-8523, Japan. ✉email: rons2k@yahoo.co.uk

By 2100, the likely range of global temperature increase relative to 1861–1880 will be 2.0 to 4.9 °C, with a 5% chance that it will be <2 °C[1]. In the much nearer future (2030–2052), global warming is likely to reach 1.5 °C above pre-industrial levels if it continues to increase at the current rate; estimated anthropogenic global warming is currently increasing at 0.2 °C per decade as a result of past and ongoing emissions[2]. This is alarming, because, as temperatures increase, extreme heat events such as heat waves and drought are also projected to increase in frequency and severity[3], endangering human lives[4], affecting ecosystems[5], impacting crop yields and global food production[6], and wreaking havoc on infrastructure resulting in economic losses[7].

Against the backdrop of rapid urbanization, especially in developing regions[8], there is also growing concern about the risks that heat poses to urban dwellers[9,10]. Generally, urban areas experience higher surface temperatures than their surrounding rural areas–a phenomenon called the urban heat island (UHI) effect[11,12]. The global urban population–currently 55% of the world's total population[8]—is particularly at risk because heat increases due to climate change, including heatwaves, and their impacts on urban areas are amplified by the UHI effect[13,14]. The challenge in ensuring the quality of urban environments and well-being of urban residents is enormous, but even more so in the years to come as the proportion of the global urban population is expected to reach 68% by 2050[8].

The UHI phenomenon is one of the most important human-induced changes to the climate of the Earth's surface[15–17]. Its most important negative consequences include increased mortality and morbidity, human discomfort, increased energy consumption and greenhouse gas emissions, and impaired air and water quality[18,19]. Increases in urban temperatures also cause heat stress in people, harming their health and impairing their well-being and productivity[20]. Thus, the UHI phenomenon not only diminishes the quality of urban ecological environments, but also affects the overall livability of urban areas and cities[11,21].

Measures to mitigate the negative impacts of heat increases and extreme heat events are needed. However, given the varying physical and socioeconomic characteristics of urban areas and cities, the effective reduction of heat-related impacts requires localized adaptation[10,22]. Today, local-level heat risk assessments, including assessments of human vulnerability and exposure to heat, are at the core of this whole issue. In general, climate-related risk assessments at the local level are performed so that a precise characterization of who and what is at risk from which climate hazard, and why, can be obtained. This is an important step toward the revolution in understanding that is needed to help achieve a climate-resilient society[23]. More particularly, this knowledge generation can help make the risks that societies and economies face visible and fully understood[23]—a crucial factor in the identification of appropriate adaptation options, which is an integral part of the adaptation planning process[24,25].

The scientific literature on climate-related vulnerability shows that the vulnerability assessment framework of the Intergovernmental Panel on Climate Change (IPCC) in its Third and Fourth Assessment Reports (TAR and AR4, respectively)[26,27] has been widely used[28]. In this framework, the vulnerability of a system to climatic stimuli is expressed as a function of its exposure, sensitivity, and adaptive capacity[26,27]. However, in its Special Report on Managing the Risks of Extreme Events and Disasters to Advance Climate Change Adaptation[29] and its Fifth Assessment Report (AR5)[30], the IPCC turned its focus on to a risk-centered assessment framework, in which risk is expressed as a function of three components, namely hazard, exposure, and vulnerability. At least at the component level, the IPCC's revised framework is consistent with the risk framework in the field of disaster risk

reduction and management[31], which is also reflective of the widely known Crichton risk triangle[32].

Located in Southeast Asia, the Philippines is ranked third worldwide in terms of disaster risk in the World Risk Index 2018, a composite index that also includes risks from climate-related hazards (www.WorldRiskReport.org). As of 2015, as many as 40.5% of the country's total population (100.98 million) were city dwellers[33]. Today, heat health risk is considered one of the key risks in the country, and thus adaptation to heat-related health impacts is one of the nation's adaptation priorities[34]. However, the literature on heat health risk and vulnerability assessments in the country is still limited. Previous heat risk-related studies in some cities and metropolitan regions of the country have come from the field of public health[35,36], but they were not necessarily conceptualized according to the IPCC AR5's risk framework. Other studies have focused on perceived heat stress in urban Philippines and how heat stress has affected intentions to move as an adaptation strategy[20,37].

Advances in remote sensing technology (e.g., remotely sensed thermal data) have been helpful, not only in the study of the UHI phenomenon[12,21,38], but also in the assessment of heat health risk and vulnerability[39–45]. Here, drawing upon the concept of risk as a function of hazard, exposure, and vulnerability, and the latter as a function of sensitivity and capacity[30,46], we assessed the current heat health risk in 139 (out of 145) Philippine cities during the hot dry season by using remote sensing data and social-ecological indicators. We also discuss the implications of our findings for adaptation planning. With its scope and approach, to our knowledge, our study is the first of its kind in the country.

The cities at high or very high risk are found in Metro Manila, where levels of heat hazard and exposure are high. The most vulnerable cities are, however, found mainly outside the national capital region, where sensitivity is higher and capacity to cope and adapt is lower. Cities with high levels of heat vulnerability and exposure must be prioritized for adaptation. This study, in general, should be useful in the context of community-level risk profiling, which is an important climate risk management-related strategy in the country[47]. The results will also contribute to the understanding of city-level heat health risks in developing regions of the Asia-Pacific.

## Results
**Minimum temperature thresholds for heat health risk.** From the remotely sensed land surface temperature (LST) data, coupled with the mortality data (see Methods), we found a minimum mortality temperature (MMT) threshold of 38.3 °C for daytime (Fig. 1a) and 24.3 °C for nighttime (Fig. 1b). For both daytime and nighttime, the relative risk (RR) curve showed an increasing trend as temperature increased beyond the MMT. The trend of the RR curve on the left side of the MMT for daytime was similar to that for nighttime: both curves first increased as the temperature decreased below the MMT before individually approaching the null RR (= 1) at a certain temperature (see Fig. 1). The observed increasing trend in the RR curve at nighttime as temperature increased beyond the MMT indicated that heat health risk was apparent even at night, contrary to the popular public view that risk is only present during the day owing to human exposure to daytime temperatures. UHI effects are present not only during the day, but also at night, as the heat accumulated in the daytime is released during the night[15,42]. Therefore, considering the importance of both nighttime and daytime temperatures in UHI studies and heat health risk assessments, we used both nighttime and daytime temperatures to produce overall heat hazard indexes for Philippine cities, taking into account the derived MMT thresholds (see Methods).

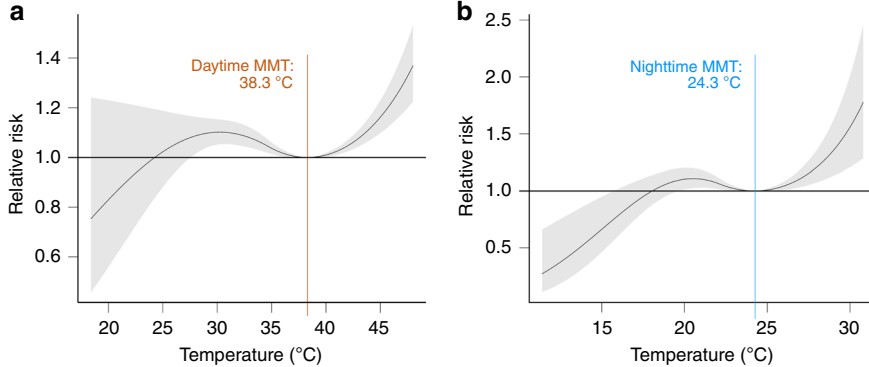

**Fig. 1 Relative risk (RR) curves derived in this study.** The RR curves show the minimum mortality temperature (MMT) thresholds for **a** daytime and **b** nighttime. The MMT thresholds were used in the preparation of a heat hazard layer. MMT refers to the temperature at which the probability of dying is lowest. RR refers to the probability of a person dying when exposed to non-optimum temperature as a ratio of the probability of a person dying when exposed to optimum temperature (i.e., the MMT). See Methods for details.

**Heat hazard in Philippine cities.** From the remotely sensed daytime and nighttime LST data during the hot dry season of c. 2015 (March–May 2014–2016) (see Fig. 2 and Methods) and the derived MMT thresholds for daytime and nighttime (Fig. 1), 11 (7.9%) of the 139 cities that were considered in the analysis emerged as having very high heat hazard index values (Fig. 3a). All these cities except one (Bacoor City, located south of Metro Manila) are in Metro Manila. A total of 19 (13.7%) cities had high heat hazard index values, four of which were in Metro Manila and the rest were distributed mostly in Luzon. The other cities fell into the moderate (26, 18.7%), low (30, 21.6%), or very low (53, 38.1%) heat hazard index categories. Cebu City and Davao City, which are regional centers in Visayas and Mindanao (Fig. 2a), respectively, both fell into the very low category. Overall, these results are evidence of the UHI phenomenon, in which the cities with higher levels of heat hazard (high or very high) were those found in the prime urban center of the country, Metro Manila. These cities have greater proportions of built-up and impervious surfaces and less vegetation and open spaces. The whole of Metro Manila itself is also ranked high in the world in terms of anthropogenic heat, which is also an important factor influencing UHI effects[48].

**Heat exposure in Philippine cities.** On the basis of population density (see Methods), only one city had a very high heat exposure index, and this was Manila City (Fig. 3b). In fact, as of 2015, Manila City, with a population density of 42,628 people per km[2] [33], was the most densely populated city, not only in the country but also in the world. Two cities had high heat exposure index values; these were Caloocan City and Mandaluyong City, both of which are in Metro Manila. The six cities with moderate heat exposure index values were also located in Metro Manila. Together, these cities classified as having moderate, high, or very high heat exposure index values accounted for 6.5% of the 139 cities. A large proportion of cities fell into the very low heat exposure index category (121, 87.1%); only nine (6.5%) cities were classified into the low index category. The least densely populated city in the country is Puerto Princesa City in the province of Palawan (Luzon), with 107 people per km[2]. The regional centers in Visayas (Cebu City) and Mindanao (Davao City) (Fig. 2a) have population densities of 3148 and 668 people per km[2], respectively. All three of these cities had very low heat exposure index values.

**Heat vulnerability in Philippine cities.** Based on the average results of the heat vulnerability assessments (see Methods), 18

(12.9%) cities had very high heat vulnerability index values; only one of these (Malabon City) was in Metro Manila (Fig. 3c). Forty-five (32.4%) cities had high heat vulnerability index values. Seven of these cities are found in Metro Manila, whereas the rest are distributed across the country. The vulnerability index values of the cities depended on the relative weights of sensitivity and capacity (Supplementary Tables 2 and 3) and on their respective values for each of these two components of vulnerability (as influenced by their respective indicators' values and relative weights). Capacity had an average weight of 0.56, whereas sensitivity had 0.44 (Supplementary Table 2). The very high heat vulnerability index of Malabon City was heavily influenced by its very low capacity index value. Overall, however, the most vulnerable cities were found mainly outside Metro Manila and were those with high sensitivity (with high poverty incidence and proportions of young and old people) and low capacity to cope and adapt (with low city and per capita net incomes and less green space).

**Heat health risk in Philippine cities.** Of the 139 cities examined, one emerged as having a very high heat health risk index (HHRI) (0.81, with a 95% CI of 0.71–0.91): Manila City, the country's capital (Fig. 4 and Supplementary Table 4). Five (3.6%) cities had high HHRI values, and they were also located in Metro Manila. The HHRI values of the cities depended on the relative weights of hazard, exposure, and vulnerability (Supplementary Tables 2 and 3) and on their respective index values in each of these three components of risk (Fig. 3). On average, exposure had the highest relative weight (0.45), followed by vulnerability (0.33) and hazard (0.22) (Supplementary Table 2). The much higher relative weight of exposure made Manila City—the top city in terms of heat exposure index because of its very high population density (Fig. 3b)—the top city in terms of heat health risk (Fig. 4 and Supplementary Table 4). Twelve (8.6%) cities had moderate HHRI values; nine of these were in Metro Manila and included Makati City (the financial center of the country) and Quezon City (the nation's most populous city). Notably, all of the cities in Metro Manila that were considered in the analysis were among the top 20 cities in terms of HHRI (Fig. 4 and Supplementary Table 4).

There were inconsistencies in the HHRI values of the cities across the 12 assessment results (Fig. 4b and Supplementary Table 4—95% CI; see Methods). These were the result of differences in the expert relative weights that were used during the aggregation process, for example, at the risk component level (hazard: mean = 0.22 and SD = 0.23; exposure: mean = 0.45 and SD = 0.22;

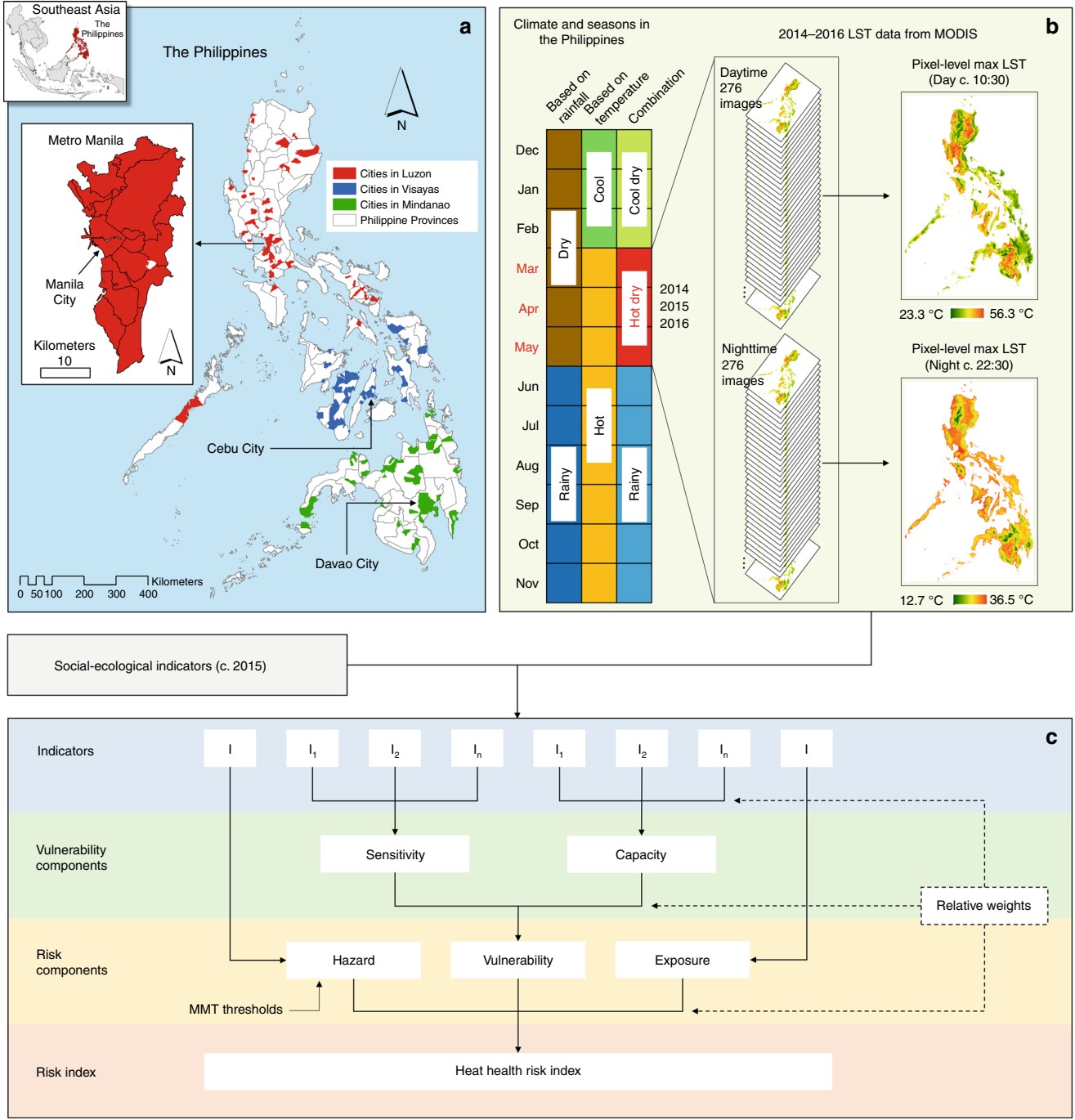

**Fig. 2 Design and framework of the study. a** Spatial distribution of cities across the three main island groups in the Philippines: Luzon, Visayas, and Mindanao. GIS data for administrative boundary layers were sourced from the Philippine GIS Data Clearinghouse (http://philgis.org). **b** Climate and seasons in the country, with the 2014–2016 satellite-derived land surface temperature (LST) data during the hot dry season (source: https://modis.gsfc.nasa.gov). The climate and seasons in the Philippines were based on data from the official site of PAGASA-DOST (http://bagong.pagasa.dost.gov.ph) and the Project SARAi's Training Toolkit–Volume 2: Climate, Weather, and Climate Change (http://sarai.ph/). **c** Diagram showing the nested hierarchical structure of the risk assessment framework employed to operationalize the IPCC's risk concept in its AR5. Supplementary Table 1 lists the factors and indicators used. See Methods for details.

vulnerability: mean = 0.33 and SD = 0.12) (Supplementary Table 2). Heat health risk assessment is multi-disciplinary and hence cuts across various fields, requiring a diversity of expertise. Therefore, although the derived weights and the risk assessment results were not consistent among experts, this result was not completely unexpected because it reflects a diversity of views from a set of experts in various, but related, fields. Nevertheless, the overall HHRI was positively and significantly correlated with the number

of deaths attributable to heat (NDAH) (Pearson's $r = 0.436$, $p < 0.0005$) (Supplementary Fig. 1; see Methods).

## Discussion

In the context of adaptation planning, although risks cannot be fully eliminated, adaptation measures should be able to reduce vulnerability and exposure, and at the same time increase

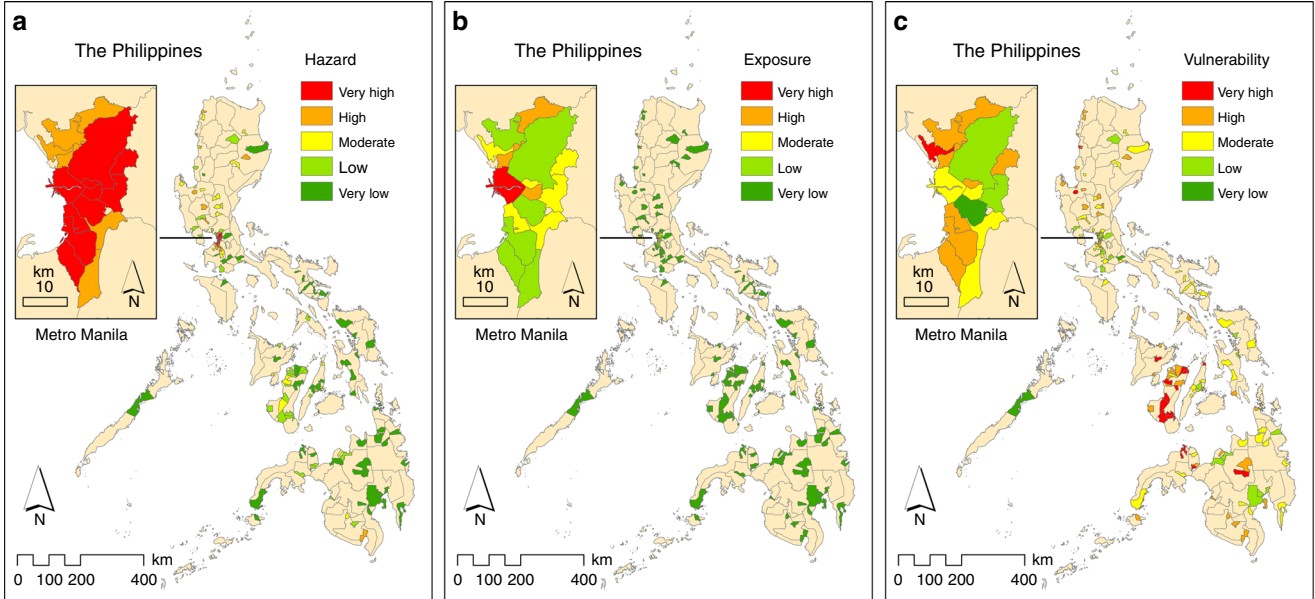

**Fig. 3 Spatial distribution of Philippine cities, with their current levels of heat hazard, exposure, and vulnerability (c. 2015).** Here, the vulnerability map represents the average of the 12 vulnerability assessments (see Methods). For hazard (**a**), the factor considered was heat—remotely sensed land surface temperature was used as an indicator. For exposure (**b**), the factor considered was exposed human population—population density was used as an indicator. For vulnerability (**c**), age structure and socioeconomic status were considered as factors for sensitivity, whereas natural resource and socioeconomic resource were considered as factors for capacity—proportion of young population, proportion of old population, and poverty incidence were used as indicators for sensitivity, weheraes availability of green space, city net income, and per capita net income were used as indicators for capacity. For details, see Supplementary Table 1.

resilience[29,30]. Here, adaptation refers to "the process of adjustment to actual or expected climate and its effects. In human systems, adaptation seeks to moderate or avoid harm or exploit beneficial opportunities. In some natural systems, human intervention may facilitate adjustment to expected climate and its effects" [p. 5][30]. Not only do adaptation options need to be location-specific, they also need to be hazard- or problem-specific. Location-specific because a particular community may have environmental and socioeconomic characteristics (including cultural and political) that are not shared by other communities in the region; thus an adaptation measure may not be generalizable across geographical regions and societal space[10,22]. Hazard- or problem-specific means that an adaptation measure should be explicit and specific concerning who is going to adapt to which climate-related hazards and impacts[49,50]. In the context of this study, it is the city dwellers themselves who need to adapt to the impact of heat so that their vulnerability, and the risks that heat poses to their health and well-being, will be lessened.

Composite indexes, such as the derived risk index in this study (Eq. (1); Fig. 4 and Supplementary Table 4), the world risk index (www.WorldRiskReport.org), the environmental performance index (https://epi.envirocenter.yale.edu/), and the sustainable development goal (SDG) index (https://sdgindex.org/), have the ability to capture the bigger picture, e.g., the multidimensionality of complex systems, such as a social-ecological system, and to provide summary statistics that can communicate system status and trends to a wide range of audiences[51]. Composite indexes are also "suitable tools whenever the primary information of an object is too complex to be handled without aggregations" [p. 13][52]. Nevertheless, the components and indicators from which such composite indexes are derived should be given more attention at the levels of planning, policy formulation, and decision making[51]. For this study, the nested hierarchical structure of the derived risk index (Fig. 1c and Supplementary Table 1) enabled us to identify the

cities where heat hazard was more intense and those cities with relatively more exposed and vulnerable populations (Fig. 3).

Indeed, the vulnerability of city dwellers to heat contributed largely to the heat health risk levels of most of the cities examined, especially those outside Metro Manila (Fig. 3c). Most of the cities outside Metro Manila have lower socioeconomic status—an important factor contributing to their vulnerability. In other studies, it has also been concluded that rural areas may be more vulnerable to heat than urban areas[53]. Our results indicate, however, that cities in a metropolitan area can also be highly vulnerable to heat (Fig. 3c)—a finding that has also been observed by other scholars[54]. Metropolitan areas around the world—but more especially in developing regions such as the Philippines—also face various issues that influence their vulnerability, including urban poverty, congestion, and poor health conditions, among other social and environmental problems. In terms of the IPCC's conceptual definition of vulnerability[30], as implemented in this study, there is a need for adaptation measures (e.g., development programs and policies that can reduce poverty incidence) that can reduce the sensitivity of people in the cities that were found to be more vulnerable to heat. Similarly, adaptation measures that can improve people's capacity to cope with, and adapt to, heat are needed, such as development programs and policies that can elevate their socioeconomic status and improve health care services.

More specifically, air-conditioning[53,55,56] can help people adapt to heat, but their economic capability will be an important factor in this case because electricity costs in the Philippines are among the highest in the world[57]. The Philippine national government, in general, needs to find means and institute measures that will lower the country's electricity costs to help its citizens to be able to adapt to heat increases, especially those in urban areas given the effects of the UHI phenomenon. That being said, the widespread use of air-conditioning is also energy-intensive and exacerbates emissions (if fossil fuel-based), leading to positive

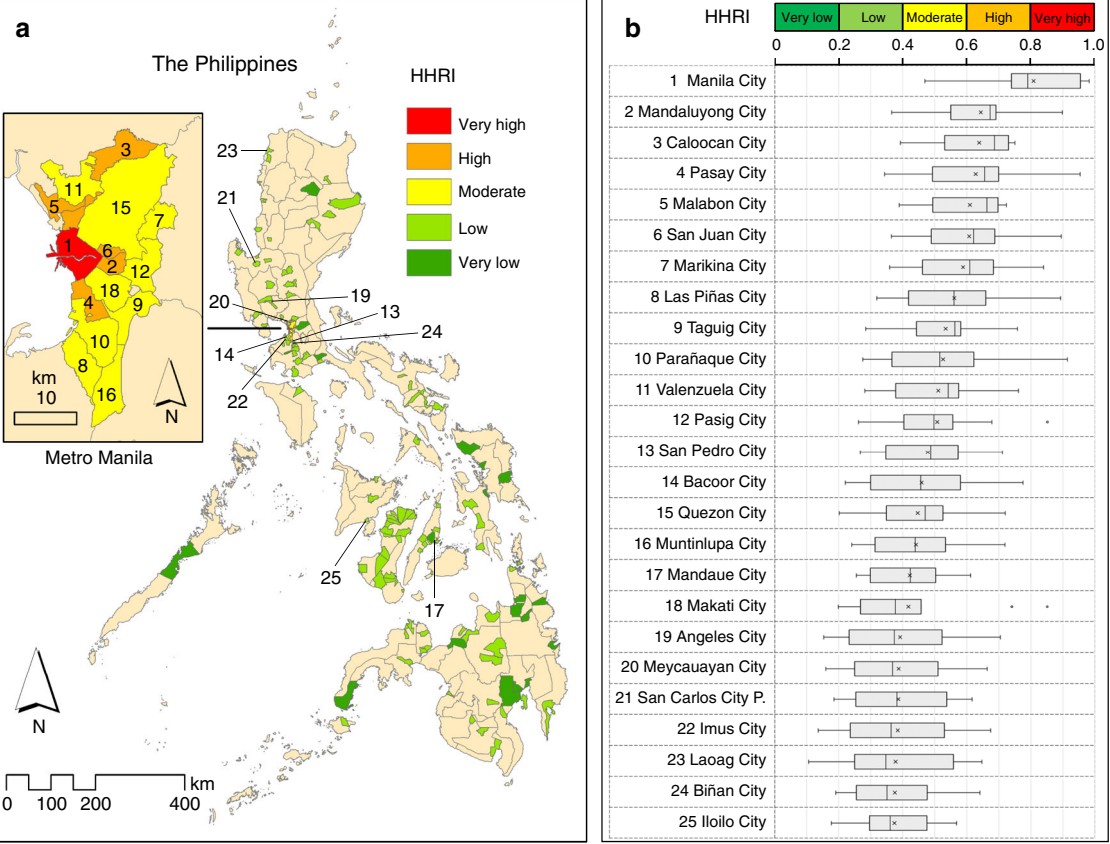

**Fig. 4 Derived current overall heat health risk index (HHRI) values for Philippine cities (c. 2015). a** Spatial distribution of the cities, with their corresponding current heat health risk levels. **b** Boxplots of the top 25 cities in terms of HHRI ($n = 12$). The central rectangle of each box plot spans from the first quartile Q1 (left side end of the box) to the third quartile Q3 (right side end of the box), which is the interquartile range (IQR=Q3-Q1)[90,91]. The vertical line inside each box is the median. In cases where the HHRI values are higher than the upper fence for outliers (Q3+1.5×IQR), the upper whisker (the end of the line that is on the right side of the box) is the highest HHRI value below the upper fence; otherwise, the upper whisker is equal to the highest HHRI value[90,91]. In cases where the HHRI values are lower than the lower fence for outliers (Q1−1.5×IQR), the lower whisker (the end of the line that is on the left side of the box) is the lowest HHRI value above the lower fence; otherwise, the lower whisker is equal to the lowest HHRI value[90,91]. The symbol "x" on the boxplots represents the average of the 12 risk assessment results, considered here as the overall HHRI of each city; these values were the ones used to produce the map (**a**). The HHRI distributions were mostly skewed, with an apparent positive skew for Manila City. The "P." in "21 San Carlos City P." refers to the province of Pangasinan. Summary results for all the cities are presented in Supplementary Table 4. See Methods for details.

feedback loops. This problem should also be considered by the government and by the country's energy, manufacturing, commercial, and household sectors.

Our findings also showed that, although vulnerability was an important contributor to risk, it was not the cities with very high levels of vulnerability (i.e., those located mostly outside of Metro Manila; Fig. 3c) that topped the list in terms of heat health risk. Instead, it was those highly densely populated cities in Metro Manila (Fig. 3b) that occupied the upper bracket of the list (Fig. 4 and Supplementary Table 4). Therefore, where possible, the level of exposure should also be lowered as part of the adaptation strategy[29,30]. In the context of this study, this means that the populations exposed to, and affected by, heat need to decrease—especially those in the highly densely populated cities of Metro Manila, like Manila City, Mandaluyong City, and Caloocan City (Figs. 3b and 4). In fact, a recent study of perceived heat stress in the urban Philippines showed that the level of heat stress increased with population density[20].

However, decongesting Metro Manila—although possible—will not be an easy task. There is a need for a national-level policy that could promote sustainable regional growth and development so that people from the provinces will no longer flock to the highly densely populated metropolitan area. If socioeconomic

opportunities in regional areas are improved, it would also become possible for some of those currently living in the metropolitan area to go back to their own provinces. In a recent survey of human mobility intentions in response to heat in urban Southeast Asia, including the urban Philippines, nearly a quarter (23%) of respondents reported that they were very likely to move away from their current locations because of heat stress, and 50% responded that they probably would[37]. This movement of people away from their current locations is an adaptation strategy[37], consistent with the characterization by the Global Commission on Adaptation of the "reduce (and prevent) and prepare (and respond)" elements of climate change adaptation[23].

Similarly, heat hazard was also a major contributor to the heat health risk levels of the cities examined, but especially those in Metro Manila (Fig. 3a). The observed pattern of heat intensity— i.e., the finding that the national urban center (Metro Manila) is hotter than its surrounding urban areas and cities (Fig. 3a)— provides evidence of the UHI phenomenon at a national scale. Complementary to the adaptation options mentioned above for reducing vulnerability and exposure to heat, measures that could lower or mitigate the intensity of heat hazard should also be considered. In this respect, various urban development-related measures (e.g., use of high-albedo materials in building design,

use of cool materials for roof, street, and pavement covers and implementation of urban greening strategies) are among the current research foci in this field[58,59].

In fact, urban greening alone—improvement of the provision, quality, and accessibility of urban green spaces—has been an important issue in the fields of environmental health, urban quality of life, urban ecology, and ecosystem-based adaptation in cities[60–62]. This is because urban green spaces provide valuable ecosystem services, including the regulation of micro-climates and urban temperatures and the purification of air[63,64]. Therefore, should there be a plan for gentrification and urban renewal in the near future in any of the cities examined, the concept of urban greening should form part of their respective development plans. One local example is the Bonifacio Global City (https://bgc.com.ph/) located in Taguig City, Metro Manila, where there have been efforts to take urban green space into account in urban planning and development.

Our overall findings on heat health risk show that cities located in the national capital region (Metro Manila) are more at risk from heat than cities outside of the country's prime urban center. Other studies have also found that heat health risk is generally higher in urban areas than in rural areas[41,43,53,54]. Although the correlation between the derived HHRI and the NDAH was not very high ($r = 0.436$), the relationship was statistically significant ($p < 0.0005$) (Supplementary Fig. 1). Some factors that could have influenced this level of correlation include the coarseness of the mortality data used (province level, instead of city level), the temporal inconsistency between the mortality data (c. 2010) and the HHRI (c. 2015), cloud contamination of the remote sensing data, which affected the level of completeness of the datasets used, and the omission of other potentially relevant vulnerability indicators owing to data limitation (more on this below). Nevertheless, the validation results are indicative of the usefulness and potential of the risk assessment approach employed, as well as of the indicators used. Empirically, our results are important and should be useful in the context of community-level risk profiling, which is much needed to support adaptation planning in the country, considering that adaptation to heat-related health impacts is one of the national adaptation priorities[34,47].

Methodologically, our study attempted to introduce some advances in the field of risk assessment. For instance, it attempted to operationalize the latest climate-related impact assessment framework by the IPCC in its AR5 (i.e., the risk framework) (Fig. 2c). In AR5, exposure refers to the presence of specific exposed elements or elements at risk (e.g., people, infrastructure, and ecosystems)[30], whereas exposure in the TAR/AR4 is a hazard-centered concept, as also reflected in various empirical studies[45,65]. Consequently, under the TAR/AR4 framing, where exposure is a component of vulnerability, vulnerability assessments suffer from uncertainties that are derived not only from the various social-ecological input data but also from the estimations or projections of climate-related hazards[50]. However, with the re-definition of exposure and its separation from vulnerability in the AR5's risk framework (Fig. 2c), vulnerability assessments no longer suffer from the same uncertainties, thereby narrowing the scope for maladaptation[50]. Some adaptation-related strategies such as green spaces[60,61] under the concept of ecosystem-based adaptation[66] also function as mitigation strategies by, for example, serving as carbon sinks. In this way, they can also help to mitigate hazard in the context of the AR5 risk framework and exposure in the context of the TAR/AR4 vulnerability framework.

In addition, this study has attempted to integrate some public and environmental health concepts (e.g., the concepts of RR and MMT) into the indicator-based assessment of heat health risk[39–43,53,54]. Such integration was vital to our study, because the derived RR curves and MMT thresholds provided the needed

critical values (temperatures) for heat hazard index determination. Other studies have considered critical temperatures based on the definition of hot days or nights or heat waves[53,54]. To our knowledge, our study is the first to integrate the concepts of RR and MMT into an indicator-based risk assessment approach, coupled with the use of remote sensing technology. The improvement in the availability and accessibility of remotely sensed thermal data has been an important technological advancement, providing alternative data for heat health risk- and vulnerability-related studies in urban areas and cities, especially in the developing regions, where data on air temperature are scarce. Remotely sensed thermal data have an important advantage, because they have a wide, contiguous spatial coverage, unlike in the case of in situ temperature data from weather stations[11,67].

Our approach also employed relative weights based on expert judgements[28,68,69], which can be either further refined in future on the basis of new knowledge on heat-related risk issues, or modified according to specific user requirements[40]. In index development involving the aggregation of various indicators, there can be a diversity of views concerning the appropriate weights and the methods used to determine them[68,69]. In this study, there were differences among expert judgements regarding the relative weights of the risk components and the components of vulnerability and their respective indicators, reflecting diversity of views (Supplementary Tables 2 and 3). As heat health risk assessment is multi-disciplinary, we attempted to capture this diversity of views by considering all the 12 assessment results in the derivation of the overall HHRI values of the cities (Fig. 4 and Supplementary Table 4; see Methods).

Our heat risk assessment approach (Fig. 2c; see Methods) also had several limitations, and these could have been a source of uncertainties in the results. In general, these limitations relate to the unavailability of detailed data (e.g., city-level data on heat-related mortality or hospital visits during the hot dry season, c. 2015). Similarly, detailed data are needed for other potentially relevant indicators for heat vulnerability assessment: sensitivity (e.g., data on pre-existing health conditions[55,56]) and capacity (e.g., data on the proportion of budgets allotted to healthcare services and environmental protection; air-conditioner use[53,55,56]; and education or literacy rates[53,55,56]). Air-conditioner use and literacy rates in the Philippines are largely dependent on the socioeconomic status of each family or individual. Therefore, in light of this local knowledge, per capita net income has been used as a proxy indicator for these variables. Nevertheless, all of these indicators mentioned, including those that are deemed relevant but have not been mentioned, need to be explored and, where possible, included in future updates of this study once the needed data at a city level become available.

Furthermore, remotely sensed thermal data support spatially explicit heat-related risk assessments at the pixel or grid level[41,43,53]. However, this study could not take full advantage of this potential of the remote sensing data that were used, owing to the lack of spatially explicit data for most of the vulnerability indicators (Supplementary Table 1)—a limitation that has been pointed out by other scholars[42]. The dynamics of the components of risk were also not considered in this study. For risk-reduction and adaptation strategies to be effective, the dynamics and interconnections of the risk components and their factors and indicators need to be considered[70]. In this respect, the climate impact chain concept—a general representation of how a given climate stimulus propagates through a system of interest via the direct and indirect impacts it entails[71,72]—as well as the introduction of different development trends, pathways, and scenarios (including future climate scenarios)[73], needs to be explored in the future.

In summary, the above discussion focuses on the potential adaptation measures that could directly address the heat exposure and vulnerability factors and indicators used in the assessment of heat health risk in Philippine cities: measures to lower heat exposure levels (regional development as a strategy to decongest Metropolitan cities) and measures to lower heat vulnerability levels (poverty reduction, socioeconomic status improvement, enhancement of healthcare services, urban greening, and lowering of electricity costs to enable wide use of air-conditioning, taking into account potential emissions). There could be other potential adaptation options, but all of the above-mentioned options can be considered in landscape and urban development planning for Philippine cities to decrease their levels of heat health risk. These measures, if implemented, could also help pave the way for the cities to become climate-resilient and steer their respective urban development toward sustainability, consistent with the SDGs (e.g., SDG 11—Sustainable Cities and Communities) (https://sustainabledevelopment.un.org/sdgs). In general, our approach is flexible and can be applied in other countries and case study areas, taking into account the caveats and limitations discussed above. Finally, we recognize that, to help advance scientific knowledge and improve understanding in the field of heat health risk, there is still a need for more local-level assessments in developing regions, which are at the forefront of the threat of the combined effects of the UHI phenomenon because of rapid, poorly planned, and spontaneous urbanization and heat increases due to climate change. This study aims to help fill and narrow these critical gaps in knowledge.

## Methods

**Study area.** The Philippines is a tropical, archipelagic country located in Southeast Asia (Fig. 2a). With a population of 100.98 million in 2015[33], it is the second most populous country in the region, next to Indonesia. Its >7000 islands are grouped into three: Luzon (north), Visayas (middle), and Mindanao (south). As of 2015, the country had a total of 145 cities, of which 73 (50.3%) are in Luzon, 39 (26.9%) in Visayas, and 33 (22.8%) in Mindanao (Fig. 2a). Metro Manila, the country's national capital region, is located in Luzon and is composed of 16 cities and one municipality. According to the 2015 census[33], the two most populous cities in the country are in Metro Manila: Quezon City (2.94 million), the country's former capital[74], and Manila City (1.78 million), the current capital.

Although the climate of the Philippines is classified into four types according to rainfall distribution (http://bagong.pagasa.dost.gov.ph/), it is generally characterized by two pronounced seasons according to the amount of rainfall: a rainy season during June–November and a dry season during December–May (Fig. 2b). Based on temperature, it is also classified into two seasons: a cool season during December–February and a hot season during March–November. The dry season can be subdivided further into a cool dry season (December–February) and a hot dry season (March–May). We focused on the hot dry season (Fig. 2b). Of the 145 cities, 139 had complete datasets and were thus considered in this study (see Supplementary Table 4).

**Heat health risk assessment.** Our study builds upon other previous related studies that have employed remote sensing and social-ecological data[39–43]. Our heat health risk assessment approach (Fig. 2c and Supplementary Table 1) implemented the IPCC's conceptual framework on risk in its AR5 in which risk is a function of hazard, exposure, and vulnerability[30]. Hazard refers to "the potential occurrence of a natural or human-induced physical event or trend or physical impact that may cause loss of life, injury, or other health impacts, as well as damage and loss to property, infrastructure, livelihoods, service provision, ecosystems, and environmental resources" [p. 5][30]. Exposure refers to "the presence of people, livelihoods, species or ecosystems, environmental functions, services, and resources, infrastructure, or economic, social, or cultural assets in places and settings that could be adversely affected" [p. 5][30]. Vulnerability refers to "the propensity or predisposition to be adversely affected [and] encompasses a variety of concepts and elements, including sensitivity or susceptibility to harm and lack of capacity to cope and adapt" [p. 5][30].

Overall, our assessment of heat health risk started with the determination of the MMT values for nighttime and daytime during the hot dry season (March–May). These values were then used as thresholds in preparing a heat hazard layer based on remotely sensed thermal data (see below). This was followed by the derivation of exposure and vulnerability layers, and the final aggregation of risk components to produce a heat health risk index for Philippine cities. The overall HHRI values of

the cities were derived by taking the averages of the 12 risk assessment results (see below); this was followed by a validation and sensitivity analysis.

More specifically, our approach is based upon the principles of an indicator-based assessment technique that is used to derive a composite index[28,68,69]. It included the identification of factors and indicators for the three risk components (hazard, exposure, and vulnerability); this identification was aided by a literature review and expert consultations. The final list of factors and indicators (Supplementary Table 1) was, however, affected by data availability. The three components of risk, as well as the two components of vulnerability (sensitivity and capacity), and their respective indicators, were aggregated by using relative weights (Fig. 2c and Supplementary Tables 2 and 3).

The relative weights were determined based on an analytic hierarchy process pairwise comparison[75] survey questionnaire administered to a number of experts (academics and researchers). In total, 26 experts were consulted (24 questionnaires were received in response; 12 of these were screened out) (Supplementary Tables 2 and 3); this number is within the range considered in other, related studies[76–78]. The concept of aggregating components and indicators and using relative weights was consistent with the IPCC's statement regarding the derivation of vulnerability indexes, namely, "a climate vulnerability index is typically derived by combining, with or without weighting, several indicators assumed to represent vulnerability" [p. 1775][46].

Before aggregation, the indicators' data values were first normalized to a common 0 to 1 value range to make them comparable to each other and enable aggregation at the indicator and component levels. Here, we used the min–max normalization method[54,68,72,79,80], a technique that performs a linear transformation of the original data. All the indicators used fell under either an interval or ratio scale of measurement (Supplementary Table 1); therefore, the same normalization method was applied to all indicators.

The risk index (RI) of city $i$ was calculated on the basis of the derived indexes of the three components of risk (see below), by using a weighted arithmetic (additive) aggregation procedure (Eq. (1))[68,72].

$$\mathrm{RI}_i = \sum_{j=1}^{n} x_{ij} w_j, \qquad (1)$$

where $x_{ij}$ is the value of city $i$ for index $j$, $w_j$ is the relative weight of index $j$, and $n$ is the number of indexes. Here, $n = 3$, referring to the three components of risk. The relative weights of the three components sum to 1 (Supplementary Table 2).

With the use of 12 sets of relative weights derived from the 12 experts' questionnaire responses (Supplementary Tables 2 and 3), the whole process produced 12 risk assessment results. The overall HHRI values of the cities were derived by taking the average of these 12 risk assessment results $\left(\frac{1}{12}\sum RI\right)$. The HHRI values were later categorized into a five-level qualitative scale: very high (0.80 to 1.00), high (0.60 to 0.80), moderate (0.40 to 0.60), low (0.20 to 0.40), and very low (0.00 to 0.20). The upper limit of each category level was inclusive.

**Minimum temperature thresholds for heat health risk.** Before proceeding with derivation of the heat hazard index, we first determined the RR curves and the MMT thresholds for daytime and nighttime (Eq. (2)). In temperature–health studies[81], RR is defined as the probability of a person dying when exposed to non-optimum temperature as a ratio of the probability of a person dying when exposed to optimum temperature (i.e., the MMT). The MMT is defined as the temperature at which the probability of dying is lowest.

$$Y_{t,c} \sim \text{Quasipoisson}$$
$$E\left(\log\left(Y_{t,c}\right)\right) = \alpha + \text{cbTemp}_{t,c,v} + \sum_{i=j}^{J} \beta_j \text{cov}_j + \text{ns}(\text{RH}_{\text{ave}}, 4) \qquad (2)$$

The daily mortality ($Y_{t,c}$) of province $c$ is assumed to follow a Poisson distribution considering overdispersion; hence the quasipoisson. The expected log of the daily mortality in each province ($E(\log(Y_{t,c}))$) was regressed with the intercept ($\alpha$), a crossbasis term (cbTemp$_{t,c,v}$) to account for the bi-dimensional aspects (lag and exposure) of temperature (for either minimum or maximum; Temp$_v$), a linear adjustment for the covariates ($\sum_{i=j}^{J} \beta_j \text{cov}_j$) of secular trends, day of the week, and holidays, and a non-linear adjustment of relative humidity (ns(RH$_{\text{ave}}$, 4)) with 4 degrees of freedom (df). The crossbasis term was parameterized by using a natural cubic spline with 4 df in the exposure dimension, an intercept with three internal knots equally spaced in the log scale of the lag dimension, and a lag period of 14 days[35,82,83]. A sensitivity analysis of the number of lag days is given in Supplementary Fig. 2. Beta estimates extracted from cbTemp$_{t,c,v}$ were used to derive the RR, which was determined by taking the exponential of the beta values. The RR was based on the 99th percentile of the temperature data.

The relationship between remotely sensed LST and measured air temperature is complicated, and it is often explored by using techniques such as statistical regression, solar zenith angle models, or thermodynamics[39,84,85]. However, in the UHI context in which this study is framed, "it is reasonable to believe that spatial trends will be similar when comparing LST and air temperature" [p. 4][39], and hence remotely sensed surface temperatures are a useful dataset, as has been shown

in previous studies of heat health risk and vulnerability[39–45]. Therefore, in drawing upon these previous studies and in response to a lack of the measured air temperature data needed to support a nationwide city-level analysis, we used a remotely sensed LST dataset to derive the RR curves and MMT thresholds.

More specifically, we used the March–May 2006–2011 LST data from the moderate resolution imaging spectroradiometer (MODIS) product distributed as MOD11A1.006 by NASA (the National Aeronautics and Space Administration)[86]. The 2006–2011 period was chosen on the basis of temporal consistency with the available mortality data, which were acquired from the Philippine Statistics Authority. The mortality data were at the province level (32 provinces) and included daily all-cause mortality records from 2006 to 2011. To be consistent, we only used the mortality data for the hot dry season (March–May).

The daily Level 3 LST product has a spatial resolution of 0.93 km, projected to a sinusoidal projection. The scientific datasets found in each daily data file (.hdf) included, among other data layers, a daytime LST raster layer (c. 10:30 local solar time) and a nighttime LST raster layer (c. 22:30 local solar time)[86,87]. We extracted the LST raster layers from each daily .hdf file, yielding in a total of 4416 raster layers (552 for daytime plus 552 for nighttime, multiplied by 4, which is the number of scenes that could cover the country). The LST raster layers contained valid values expressed in degrees kelvin ranging from 7500 to 65,535 at a scale factor of 0.02 and a fill value of 0 for pixels that had no data[87]. As part of the geoprocessing procedure, we first stacked all the LST raster layers for each scene. This was followed by a mosaicking procedure that resulted into two mosaics of stacked layers– one for daytime and one for nighttime. We then extracted each mosaicked LST raster layer before converting the LST values to degrees Celsius (°C).

Finally, by using a geoprocessing technique called zonal statistics–a tool in ArcMap that calculates statistics on values of a raster within the zones of another dataset–we determined the maximum daily daytime and nighttime LSTs in each of the provinces with mortality data, considering all the pixels covered by each province. The derived maximum daily daytime and nighttime LSTs over the 552-day period for the 32 provinces with mortality data were used as input variables to Eq. (2). Supplementary Fig. 3 is a flow diagram of the main steps implemented in the processing of the MODIS LST data to derive the RR curves and MMT thresholds. There were days on which the pixels covered by a province had no data owing to the presence of clouds; this resulted in some provinces having no data for maximum LST on some days (see also Supplementary Fig. 4). We filled in the missing maximum LST values for those days in such provinces by employing a multivariate imputation with the use of chained equations, implemented in the R programming statistical package called mice[88].

**Deriving a heat hazard index.** By drawing upon previous studies[39–43], we also used remotely sensed LST as a proxy indicator for heat. As in the above-described derivation of the RR curves and MMT thresholds, we also used MODIS LST data (MOD11A1.006) to derive the heat hazard index. We focused on the current (c. 2015) heat health risk in Philippine cities during the hot dry season (Fig. 2b). The average maximum temperature for the whole hot dry season (March–May) is generally higher than for the whole hot rainy season (June-August) (Supplementary Fig. 5). Furthermore, although there could still be a heat health risk during the hot rainy season, our province-level analysis using available air temperature data revealed that the relative risk from heat was higher during the hot dry season (Supplementary Fig. 6). During the rainy season, the degree of cloud contamination of remotely sensed LST data is also higher, posing challenges to inter-seasonal analysis. In a separate study in Zhejiang Province, China, 3-month study periods were also used[53]. Nevertheless, while we focused only on the hot dry season, we added two more years to the March–May 2015 dataset (i.e., March–May 2014 and March–May 2016) to capture the natural inter-annual variation in LST (Fig. 2b). We used the period 2014–2016 so that the resulting heat hazard layer from the LST data would remain temporally consistent with the social-ecological datasets used for the exposure and vulnerability layers (c. 2015) (Fig. 2 and Supplementary Table 1).

We extracted the LST raster layers from each daily .hdf file; this resulted in a total of 2208 raster layers (276 for daytime plus 276 for nighttime, multiplied by 4 scenes). There were also pixels that did not have valid values on some days because of cloud cover (i.e., no data) (see Supplementary Fig. 7). This hindered us in using the number of days with temperatures higher than the MMT thresholds as the basis for deriving the heat hazard index. As an alternative, scene by scene, we determined the maximum LST of each pixel over the 276-day period, excluding the pixels that had no data. This resulted in eight final scenes: four for daytime and four for nighttime. We mosaicked the four scenes for daytime and the four scenes for nighttime to produce two composite LST maps for the whole country: one for daytime and one for nighttime (Fig. 2b and Supplementary Fig. 8). We then converted the LST values to °C.

Before the two LST maps were normalized into a 0 to 1 index-value range, the pixels that were within the boundaries of the cities were first extracted. In the normalization process, we used the derived MMT values as thresholds for daytime (38.3 °C) and nighttime (24.3 °C) (Fig. 1), meaning that pixels with LST ≤ MMT for daytime and ≤ MMT for nighttime would have the lowest heat hazard index value, i.e., 0 in the normalized value range of 0 to 1. The heat hazard index increased from 0 to 1 as the LST of the pixels increased from the MMT threshold to the highest

LST. After normalization, the heat hazard index maps (daytime and nighttime) were averaged. The overall heat hazard index of each city was determined from this averaged heat hazard index map by taking the mean of the index values of the pixels inside each city. Finally, the resulting heat hazard index values of the cities were normalized to a 0 to 1 index-value range and categorized into a five-level qualitative scale, as for HHRI. Supplementary Fig. 8 is a flow diagram of the main steps implemented in processing the MODIS LST data to derive the heat hazard index.

**Deriving a heat exposure index.** By drawing upon the IPCC's concept of exposure[30], as explained above, we defined exposure in the context of this study as an index referring to the presence of people that could be adversely affected by heat. In this regard, we used population density as an indicator of heat exposure (Supplementary Table 1). This indicator has been used in other heat health risk and vulnerability studies[39,40,42,44]. To be consistent with the way the heat hazard index was produced, we first derived a spatially explicit heat exposure index before determining the overall heat exposure index of each city.

In the absence of spatially explicit population distributions, previous studies have opted to use a proxy index, called the elevation-adjusted human settlement index, which can be produced by using a set of remotely sensed data, including a nighttime lights dataset, a vegetation index, and a digital elevation model[41,43,53]. Fortunately, the downscaling of population data has progressed over the past years. Global gridded population data are now available at various spatial resolutions and time points. For instance, the gridded population data products of WorldPop for several corresponding years have a spatial resolution of 100 m (www.worldpop.org)[89]. We took advantage of the availability of this dataset and used the spatially explicit 2015 population density map of the Philippines. The scatter plot between total city population based on this dataset and total city population based on the 2015 census had a very strong (nearly 1:1) positive correlation ($R^2 = 0.9926$) (Supplementary Fig. 9).

As in the case of the heat hazard index, we first extracted the pixels of the population density map that were within the city boundaries. We then normalized the extracted pixels containing population density values to the same 0 to 1 index-value range. We assumed that the heat exposure index value increased from 0 to 1 as the population density increased from the lowest to the highest value. The overall heat exposure index of each city was derived by taking the average of the index values of the pixels inside each city. Finally, the overall heat exposure index values of the cities were also normalized to a 0 to 1 index-value range and categorized into the same five-level qualitative scale as described above.

**Deriving a heat vulnerability index.** In the IPCC AR5, vulnerability is a function of sensitivity and capacity[30,72] (Fig. 2c). Considering data availability, the final factors for sensitivity included age structure (as indicated by the proportion of the young population (<15 years) and the proportion of the old population (≥65 years)), and socioeconomic status (as indicated by poverty incidence) (Supplementary Table 1). For capacity, the final factors were natural resource (as indicated by green space) and socioeconomic resource (as indicated by city net income and per capita net income) (Supplementary Table 1). With the exception of green space, which was indicated by a vegetation index called the enhanced vegetation index (EVI), captured at 16-day interval (MOD13Q1) (Supplementary Table 1), all of these indicators were at the city level and were non-spatial.

For the indicator green space, the index value of each city was derived by first calculating the average EVI of each pixel over the hot dry season (March–May 2015). This was followed by a mosaicking process for the four scenes that covered the whole country and that contained pixels with average EVI values. The same zonal statistics tool, as described above, was used to derive the mean EVI of each city. We normalized all the indicators to the same 0 to 1 index-value range, considering the following assumptions. First, the higher the proportions of young and old population and the higher the incidence of poverty, the more sensitive the city population is to heat. Second, the higher the mean EVI, the city net income, and the per capita net income, the higher the capacity of the city population is to cope and adapt to heat.

Subsequently, the indicators were aggregated in accordance with the logic of Eq. (1) to produce a sensitivity index and a capacity index. The same aggregation technique was used to produce a heat vulnerability index, composed of these two indexes. Before aggregation, however, the capacity index was first inverted because of its inverse relationship with vulnerability. The 12 sets of relative weights (Supplementary Table 2) were used during aggregation of the indicators and the two vulnerability components. Finally, the vulnerability index values of the cities from each of the 12 heat vulnerability assessment results were also normalized to a 0 to 1 index-value range.

**Validation and sensitivity analysis.** Plotting the derived HHRI values of the cities against an observed heat health-related impacts (e.g., recorded heat-related mortality[53] or summer hospital visits[43]) can provide a direct validation of the risk assessment results. Here, because of a lack of data at the city level for the year c. 2015, we used the available province-level all-cause mortality data for the hot dry season of 2009–2011. The 32 provinces with all-cause mortality data (Supplementary Fig. 4a) altogether covered 65 cities. To estimate all-cause mortality at the

city level, we calculated the ratio of a city's population (2010) to the total population of the province (2010) in which it was located. This ratio was used to estimate the city's share of all-cause mortality ($M$) from the province's average all-cause mortality during the period. By using Eqs. (3) and (4), we estimated the city-level NDAH, expressed as a density at city level ($km^{-2}$). The use of density at city level is consistent with the heat exposure index, which was produced by using population density (see above). Finally, a scatter plot was produced and the correlation (Pearson's $r$) between HHRI and NDAH was calculated (see Supplementary Fig. 1).

$$\text{NDAH}_{city}(km^{-2}) = x/A \qquad (3)$$

$$x = ((RR - 1)/RR) \times M \qquad (4)$$

where $x$ is the number of deaths attributable to heat at city level, $((RR - 1)/RR)$ is the attributable fraction, interpreted as the excess risk due to heat, $M$ is the city's share of all-cause mortality from the province's all-cause mortality, and $A$ refers to the city's land area. Here, for $x$, the average between day and night was used to make the validation consistent with the way in which the heat hazard index was produced, which was also based on the average between the day and night heat hazard indexes (see above). The relative risk (RR) was derived on the basis of Eq. (2).

Additionally, we calculated the 95% confidence intervals (CIs) of the derived HHRI values of all of the cities (Supplementary Table 4). We also performed a sensitivity analysis for the number of lag days in deriving the RR curves and MMT thresholds (Supplementary Fig. 2). A comparative analysis between RR in the hot dry season and RR in the hot rainy season was also performed (Supplementary Fig. 6). Finally, the spatiotemporal completeness of the MODIS LST data used was also determined (Supplementary Figs. 4 and 7).

**Implementation.** Production of the RR curves and derivation of the MMT thresholds and the number of deaths attributable to heat ($x$), as well as calculation of the 95% CI, were accomplished by using the statistical program, R version 3.5.3. Extraction of raster LST layers from the MODIS .hdf files and the subsequent processing of the extracted data, including all the mapping activities, were performed in ArcMap 10.5. All other statistical analyses were performed by using Microsoft Office 365 ProPlus Excel Version 1902.

**Reporting summary.** Further information on research design is available in the Nature Research Reporting Summary linked to this article.

## Data availability
The sources of all the data used are acknowledged in the Methods and Supplementary Information sections. Complete lists of the cities, with their derived HHRI values, and the 12 sets of relative weights used, are given in the Supplementary Information.

## Code availability
The R scripts and ArcMap models developed and used in this study are available from the corresponding author upon request.

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

## Acknowledgements

This research was supported by the Environment Research and Technology Development Fund 2–1708 of the Environmental Restoration and Conservation Agency of Japan, the Japan Society for the Promotion of Science (JSPS), through Grant-in-Aid for Scientific Research (KAKENHI) 18H00763 (Representative: Yuji Murayama), and the Climate Change Adaptation Program of the National Institute for Environmental Studies (NIES), Japan. We also acknowledge and sincerely thank all the experts (academics and researchers) consulted for this study. This study also benefitted from R.C.E.'s discussions with Satbyul Kim and Noriko Ishizaki. The conclusions and recommendations presented in this article are of the authors and do not, in any way, represent the views of the funders or the authors' respective institutions.

## Author contributions

R.C.E. conceived and designed the study, conducted the research, and wrote the paper. M.O. helped conceive the study. M.O., T.T., and Y.H. provided research supervision and guidance to R.C.E. and helped in the interpretation of the results. X.T.S. performed the MMT analysis and helped in the validation and sensitivity analysis and in the conduct of expert consultations. K.T. and S.N. provided inputs to the study and helped in the interpretation of the results.

## Competing interests

The authors declare no competing interests.
