## [Peer Review File · Nature Communications]

Reviewers' comments:

Reviewer #1 (Remarks to the Author):

Based on the IPCC conceptual framework for risk assessment, this paper assessed and mapped the heat health risks in 139 Philippine cities in 2015. Heat hazard index, heat exposure index and heat vulnerability index were separately assessed using land surface temperature data, population data and other socio-ecological data, and then the composite heat risk index was aggregated and mapped for further decision support in preparation, adaptation and mitigation of heat risks in the urban Philippines. I have the following comments:

1. The title might not be suitable because this study only concentrated on the heat health risks of 139 Philippine cities during March-May 2015. The title should include "health" and the study period.
2. Line 339 The paper only select LST data during March-May (hot dry season) in 2015. What I do not understand is why the heat health risk assessment did not use the temperature data during June-August (or September)? Hot rainy season is also hot for people. Also, the study period of three months is too short for such a nation-wide risk zoning study, because the spatial variation of temperature (i.e., the hazard index) may change greatly every year.
3. Line 340 Sources and data quality of the temperature data and mortality data should be pointed out clearly. Did one province have one weather station? What about the percentage of missing value of meteorological temperature data and LST data? Also, can the author give the data coverage of the mortality data?
4. Line 365 The study used an analytic hierarchy process to determine the weights of three risk elements based on the expert scoring method. As shown in Fig.S4, there were big differences in 12 experts' feedbacks. Thus this method may be arbitrary and the limited number of experts may greatly influence the precision of the risk assessment. Considering the author has the mortality data, can the author validate the risk map using health data?
5. Line 392 The paper use 14 days as the maximum lag days of temperature in DLNM. However, many studies found the cold effect lasted up to 21 days. Sensitivity analysis using 21 days should be done, because the set of lag days in model may influence the MMT.
6. Line 403 The coefficients of 0.74 and 0.68 are not high enough to support the maximum and minimum temperatures to be the proxies for the daytime and nighttime temperatures. Still, using LST-MMT in hazard assessment seems to be too simple considering the complex association between LST and ambient temperature—sometimes high correlation, but sometimes not. Overall, the author need to deal with the relationships between daytime/nighttime temperatures, mean temperatures for specific hours, maximum/minimum temperatures, and LSTs very carefully.
7. Line 444 Importantly, population rather than population density should be used in exposure assessment, because the study did not map the exposure and risk at pixel level. The composite heat health risks for one city are related to total number of population of this city, but not the population density.

Overall, the results of this paper do not quite meet the impact and innovation criteria of Nature Communications. Besides, there are some above-mentioned problems of the method applied. To conclude, I recommend that the paper should not be accepted for the publication in the present form, it might be more fitful for environment or geography related journals, such as ES&T, Risk Analysis and Applied Geography.

Reviewer #2 (Remarks to the Author):

I congratulate the authors for undertaking this study. It was a joy to read and showed the effort the researchers have put in this work. This paper assesses heat risk in Philippine cities. Cities identified with high levels of heat vulnerability and exposure may be prioritized for adaptation. The results will contribute to the understanding of city-level heat risks in the developing region of the Asia-Pacific.

These novel findings will be of interest to others in the region and the wider field. The authors' use of a wide range of data, and the hazard, exposure and vulnerability framework led to a comprehensive evaluation of heat risk. I particularly liked their discussion of urban decongestion and sustainable and regional development. Additionally, besides being expensive, widespread air conditioning usage is particularly energy intensive and exacerbates emissions (if energy is fossil fuel based) leading to positive feedback loops.

Though there is always room for improvement and the authors have acknowledged some weaknesses in their approach there are a few minor clarifications that would make this work more convincing. The authors could provide some further evidence to strengthen the conclusions.

1. The choice of 12 experts and the experts rating of relative weights leads to subjectivity in final HRI ranking. A sensitivity analysis could be helpful in putting these concerns to rest. The authors could show the effects of a slight perturbation (say 5%) in experts' relative weights on the city rankings.
2. The vulnerability components encompass the domains of age, green space and income (poverty, per capita income). Other studies of heat vulnerability indices typically include some measure of health, social stratification (race / caste), household amenities, education, occupation etc. While some of these variables (such as household amenities) can be proxied by income, the authors should justify and explain the omission of others.

On a subjective note, this paper will influence thinking in the field. The approach, accounting for limitations, is flexible and can be applied in other countries and settings. Evidence from developing countries, at the forefront of this threat, is less and therefore this paper fulfills that critical gap in knowledge.

The statistical analysis used is appropriate for this work. The authors have use publicly available data and therefore this work is likely reproduceable by other researchers in the field.

Reviewer #1

Based on the IPCC conceptual framework for risk assessment, this paper assessed and mapped the heat health risks in 139 Philippine cities in 2015. Heat hazard index, heat exposure index and heat vulnerability index were separately assessed using land surface temperature data, population data and other socio-ecological data, and then the composite heat risk index was aggregated and mapped for further decision support in preparation, adaptation and mitigation of heat risks in the urban Philippines. I have the following comments:

Response: We sincerely thank Reviewer #1 for the time and effort in reviewing our manuscript. We really appreciate all the comments and suggestions. In the revised manuscript, we carefully took all of these into account. Below is our point-by-point response.

1. The title might not be suitable because this study only concentrated on the heat health risks of 139 Philippine cities during March-May 2015. The title should include “health” and the study period.

Response: Thank you for pointing this out. We agree. Hence, the title has been revised and now reads “Heat health risk assessment in Philippine cities”. For the study period, we have opted to mention it in the Abstract.

2. Line 339 The paper only select LST data during March-May (hot dry season) in 2015. What I do not understand is why the heat health risk assessment did not use the temperature data during June-August (or September)? Hot rainy season is also hot for people. Also, the study period of three months is too short for such a nation-wide risk zoning study, because the spatial variation of temperature (i.e., the hazard index) may change greatly every year.

Response: Thank you for the comment.

- We agree that inter-seasonal analysis is another way to proceed, but in this study, we focused only on the hot dry season. Our analysis comparing hot dry season and hot rainy season shows that relative risk to heat is higher during the hot dry season (see Fig. S6). Fig. S5b also shows that temperature is, on average, generally hotter during hot dry season than during hot rainy season. Additionally, the level of cloud contamination on remotely sensed LST data is higher during rainy season.

Lines 521-527 of the revised manuscript:

In this study, we focused on the current (c. 2015) heat health risk during the hot dry season in Philippine cities (Fig. 1b). The average maximum temperature for the whole hot dry season (March-May) is generally higher than for the whole hot rainy season (June-August) (Fig. S5). Furthermore, while there is still heat health risk during the hot rainy season, relative risk to heat is higher during the hot dry season (Fig. S6). During rainy season, the degree of cloud contamination of remotely sensed LST data is also higher, posing challenges to an inter-seasonal analysis.

- To capture the natural inter-annual variation in temperature, we have added 2 more years (2014 and 2016), sandwiching 2015, which is the focus year of the study representing ‘current’. Fig. S5a shows the temporal pattern of average annual temperature (hot dry season only) in Manila City. The period 2014-2016 was decided so that the resulting heat hazard index from the LST data would still be temporally consistent with the social-ecological data that were used to produce the heat exposure and vulnerability indexes (c. 2015). Fig. S8 shows the schematic diagram of the MODIS LST data processing for the derivation of the heat hazard index.

Lines 527-532 of the revised manuscript:

Nevertheless, to capture the natural inter-annual variation in LST during the hot dry season, we added two more years to the 2015 March-May dataset (i.e. 2014 March-May and 2016 March-May) (Fig. 1b). The 2014-2016 period was considered so that the resulting heat hazard layer from the LST data would still be temporally consistent with the social-ecological datasets used for the exposure and vulnerability layers (c. 2015) (Fig. 1; Table S1).

3. Line 340 Sources and data quality of the temperature data and mortality data should be pointed out clearly. Did one province have one weather station? What about the percentage of missing value of meteorological temperature data and LST data? Also, can the author give the data coverage of the mortality data?

Response: Thank you for the questions and suggestions.

- To address the issue on the air temperature – LST relationship (comment 6), we have decided to use daily LST data (2006-2011) to derive new minimum mortality temperature (MMT) thresholds.

While we could derive MMTs using air temperature data, we do not have enough data to support a nationwide city-level derivation of the current heat hazard index. Hence, this decision. As a result, both the newly derived MMTs and heat hazard index are now LST-based. We have produced two maps showing the spatiotemporal completeness of all the LST data used (see Figs. S4 and S7). In addition, more detailed descriptions of the mortality data have been added.

Lines 485-493 of the revised manuscript:

The relationship between remotely sensed land surface temperature (LST) and measured air temperature is complicated, with techniques such as statistical regression, solar zenith angle models or thermodynamics are often being used to explore the relationship^{40,95,96}. However, in the context of UHI in which this study is framed, 'it is reasonable to believe that spatial trends will be similar when comparing LST and air temperature', and therefore remotely sensed surface temperatures are a useful dataset⁴⁰, as also shown in previous studies on heat health risk and vulnerability^{40-45,69,97}. Thus, drawing upon these previous studies and due to the lack of the needed measured air temperature data that could support a nationwide city-level analysis, a remotely sensed LST dataset was used to derive the RR curves and MMT thresholds.

Lines 496-499 of the revised manuscript:

The 2006-2011 period was decided on the basis of temporal consistency with the available mortality data, acquired from the Philippine Statistics Agency. The data were at the province level (32 provinces) and included daily all-cause mortality records from 2006 to 2011.

4. Line 365 The study used an analytic hierarchy process to determine the weights of three risk elements based on the expert scoring method. As shown in Fig.S4, there were big differences in 12 experts' feedbacks. Thus this method may be arbitrary and the limited number of experts may greatly influence the precision of the risk assessment. Considering the author has the mortality data, can the author validate the risk map using health data?

Response: Thank you for the comments.

- We recognize and agree that there were differences in the feedback of the experts. To address this concern and strengthen our results, we have calculated confidence intervals (95% CI) for the resulting risk index values, showing the lower and upper bounds (Table S2). It can be noted that our heat health risk assessment is a multi-disciplinary assessment, hence cuts across various fields requiring diversity of expertise. Thus, while the derived weights and the risk assessment results were not consistent among experts, this result is not completely unexpected as it reflects a diversity of views from a set of experts in various, but related, fields.

- For the validation issue, we have validated the risk map using 2009-2011 mortality data (hot dry season) and found a correlation of $r = 0.250-0.239$ for daytime and $r = 0.262-0.314$ for nighttime, both are statistically significant ($p < 0.05$). The possible factors that could have affected the level of correlation have also been discussed. Fig. S1 shows the validation results.

Lines 198-203 of the revised manuscript:

Heat health risk assessment is a multi-disciplinary assessment, and hence cuts across various fields, requiring diversity of expertise. Thus, while the derived weights and the risk assessment results were not consistent among experts, this result is not completely unexpected as it reflects a diversity of views from a set of experts in various, but related, fields. Nevertheless, the overall HHRI was positively correlated with the number of deaths attributable to heat (NDAH), both for daytime ($r = 0.253-0.335$, $p < 0.05$) and nighttime ($r = 0.265-0.321$, $p < 0.05$) (Fig. S1).

Lines 306-312 of the revised manuscript:

Some factors that could have influenced this level of correlation include: (1) the coarseness of the mortality data used (province level, instead of city level), (2) temporal inconsistency between the mortality data (c. 2010) and the HHRI (c. 2015), (3) cloud commination on the remote sensing data, which affected the level of completeness of the datasets used, and (4) the omission of other potentially relevant vulnerability indicators due to data limitation (more on this below). Nevertheless, the validation results are indicative of the accuracy, usefulness and potential of the risk assessment approach employed, as well as the indicators used.

Lines 605-625 of the revised manuscript:

Validation and sensitivity analysis. *Plotting the derived HHRI values of the cities against an observed heat health-related impacts, e.g. recorded heat-related mortality⁵⁶ or summer hospital visits⁴⁴, can provide a direct validation of the risk assessment results. Here, due to the lack of data at the city level for the year c. 2015, we used the available province level mortality data during the hot dry season of 2009-2011. The 32 provinces with mortality data (Fig. S4a) altogether cover 65 cities. To estimate the mortality at the city level, we calculated the ratio of the city's population (2010) to the total population of the province (2010) where the city belongs. This ratio was used to estimate the city's share of mortality from the province's annual average mortality during the period. Using Eq. (3), we estimated the city level number of deaths attributable to heat (NDAH). Finally, scatter plots were produced and correlations between the HHRI and the NDAH were derived (Fig. S1).*

$$NDAH = ((RR - 1)/RR) \times M \quad (3)$$

where $((RR - 1)/RR)$ is the attributable fraction, interpreted as the excess risk due to heat, and M is the city's share of mortality from the province's mortality. The relative risk (RR) was derived based on Eq. (2).

For the sensitivity analysis, we calculated the 95% confidence interval (CI) of the derived HHRI values of all the cities (Table S2). We also performed a sensitivity analysis for the number of lag days in deriving the RR curves and MMTs (Fig. S2). A comparative analysis between the RR in hot dry season and the RR in hot rainy season was also performed (Fig. S6). Finally, the spatiotemporal completeness of the MODIS LST data used was also determined (Figs. S4 and S7).

- In addition, our study considered a total of 26 experts (24 were retrieved; 12 were screened out), a number that is within the range considered in other related studies (e.g. from 6 experts to 30 experts):

Lines 433-442 of the revised manuscript:

The aggregation of the three components of risk, as well as the two components of vulnerability (sensitivity and capacity) and their respective indicators, was performed with the use of relative weights (Fig. 1c). These relative weights were determined based on an analytic hierarchy process pairwise comparison⁸⁵ survey questionnaire administered to a number of experts (academics and researchers). In total, 26 experts were consulted (24 were retrieved; 12 were screened out) (Tables S3 and S4), a number that is within the range considered in other related studies⁸⁶⁻⁹⁰. The idea to aggregate and use relative weights is consistent with the IPCC's statement when deriving a vulnerability index, i.e. "a climate vulnerability index is typically derived by combining, with or without weighting, several indicators assumed to represent vulnerability" [p. 1775]³⁷.

5. Line 392 The paper use 14 days as the maximum lag days of temperature in DLNM. However, many studies found the cold effect lasted up to 21 days. Sensitivity analysis using 21 days should be done, because the set of lag days in model may influence the MMT.

Response: Thank you for the comments.

- We have run the model using 7, 14, 21 and 30 lag days. The results show that 14 lag days was optimal since the MMT did not change anymore for daytime temperature, whereas the MMTs for nighttime temperature were comparably similar to each other. For consistency, we used a lag of 14 days for both. Fig. S2 shows the results of the sensitivity analysis.

6. Line 403 The coefficients of 0.74 and 0.68 are not high enough to support the maximum and minimum temperatures to be the proxies for the daytime and nighttime temperatures. Still, using LST-MMT in hazard assessment seems to be too simple considering the complex association between LST and ambient temperature—sometimes high correlation, but sometimes not. Overall, the author need to deal with the relationships between daytime/nighttime temperatures, mean temperatures for specific hours, maximum/minimum temperatures, and LSTs very carefully.

Response: Thank you for the comments and suggestions.

- We agree that the relationship between air temperature and LST is not perfect/not 1:1, and we recognize that this can be a source of uncertainty in the results. To address this, we have decided to use daily LST data (2006-2011) to derive new minimum mortality temperature (MMT) thresholds. While we could derive MMTs using air temperature data, we do not have enough data to support a nationwide city-level derivation of the current heat hazard index. Hence, this decision. As a result, both the newly derived MMTs and heat hazard index are now LST-based. The period 2006-2011 for the derivation of the LST-based MMTs was decided so that it would be temporally consistent with the available mortality data (2006-2011). The new MMTs were derived for the hot dry season (see Fig. 2 and revised Methods).

Lines 485-493 of the revised manuscript:

The relationship between remotely sensed land surface temperature (LST) and measured air temperature is complicated, with techniques such as statistical regression, solar zenith angle models or thermodynamics are often being used to explore the relationship^{40,95,96}. However, in the context of UHI in which this study is framed, 'it is reasonable to believe that spatial trends will be similar when comparing LST and air temperature', and therefore remotely sensed surface temperatures are a useful dataset⁴⁰, as also shown in previous studies on heat health risk and vulnerability^{40-45,69,97}. Thus, drawing upon these previous studies and due to the lack of the needed measured air temperature data that could support a nationwide city-level analysis, a remotely sensed LST dataset was used to derive the RR curves and MMT thresholds.

7. Line 444 Importantly, population rather than population density should be used in exposure assessment, because the study did not map the exposure and risk at pixel level. The composite heat health risks for one city are related to total number of population of this city, but not the population density.

Response: Thank you for the comment. We understand the point raised.

- To address the issue, we have revised our approach in deriving the heat exposure index by first calculating the index at the pixel level based on a gridded population density data (100 m) before aggregating it at the city level. This approach is also consistent with the way the heat hazard index was produced. A sensitivity analysis revealed a nearly perfect positive relationship ($R^2 = 0.9926$) between city total population derived from census and the gridded data (see Fig. S9).

- In relation to the second sentence of the comment, we would like to clarify that the composite risk index of a city is a function, not only of its total population, or population density for that matter, but rather to the three components of risk: hazard, exposure and vulnerability index values – according to the risk concept adopted and operationalized in this study. This means that a high total population or population density (exposure) does not automatically equate to high risk.

- We also think that population density is the more appropriate indicator to be used either at the pixel or city boundary level for the following reasons: (1) an overall risk index is derived, and hence the exposure layer as indicated by the presence of people that could be adversely affected needs to be

normalized based on the respective area of the cities to enable appropriate inter-city comparison (hence population density); and (2) various heat health risk studies and other climate-related risk and vulnerability assessments that used non-pixel unit of analysis have also used population density instead of total population, e.g. British lower layer super output area or LSOA – for small area statistics (Tomlinson et al. 2011; Wolf and McGregor 2013), lowest level block groups or IRIS in French (Buscail et al. 2012) and Bangladesh unions (Quader et al. 2017). The recently developed guideline that operationalizes the IPCC risk concept has also used population density (number of farmers per km²) as one of its exposure indicators in its risk assessment example (risk of loss of agricultural livelihoods due to salinity) (GIZ, EURAC & UNU-EHS 2018). Furthermore, the World Risk Index also uses normalized population-related indicators, instead of absolute population (e.g. Susceptibility Indicators: share of a population without access to basic sanitation services; share of a population without access to basic drinking water services; and share of a population living below the minimum level of dietary energy consumption) (WRI 2019).

Overall, the results of this paper do not quite meet the impact and innovation criteria of Nature Communications. Besides, there are some above-mentioned problems of the method applied. To conclude, I recommend that the paper should not be accepted for the publication in the present form, it might be more fitful for environment or geography related journals, such as ES&T, Risk Analysis and Applied Geography.

- The paper has been revised taking all the review comments and suggestions. In addition to the revisions made based on the review comments and suggestions, we also did the following: (1) added two sub-sections in the Methods: (a) Validation and sensitivity analysis, and (b) Implementation; (2) improved the Discussion section by making it more comprehensive, but cohesive, also highlighting the scientific contributions of the study empirically and methodologically; and (3) improved the Methods section by adding more details on the data and processing techniques used to enhance the replicability or reproducibility of the study.
- That said, we respect the opinion of Reviewer #1.

Again, thank you very much for the time, effort and expertise shared. We sincerely appreciate them.

Cited References

- Buscail, C. et al. Mapping heatwave health risk at the community level for public health action. *International Journal of Health Geographics* **11**, 38 (2012).
- GIZ, EURAC & UNU-EHS. *Climate Risk Assessment for Ecosystem-based Adaptation: A guidebook for planners and practitioners*. (GIZ, Bonn, 2018).
- Quader, M.A. Assessing risks from cyclones for human lives and livelihoods in the coastal region of Bangladesh. *International Journal of Environmental Research and Public Health* **14**, 831 (2017).
- Tomlinson, C. J. et al. Including the urban heat island in spatial heat health risk assessment strategies: a case study for Birmingham, UK. *International Journal of Health Geographics* **10**, 42 (2011).
- Wolf, T. & McGregor, G. The development of a heat wave vulnerability index for London, United Kingdom. *Weather Clim. Extrem.* **1**, 59–68 (2013).
- WRI (World Risk Index). Methodological notes of the WorldRiskIndex. (2019). <https://weltrisikobericht.de/english-2/>

Reviewer #2

I congratulate the authors for undertaking this study. It was a joy to read and showed the effort the researchers have put in this work. This paper assesses heat risk in Philippine cities. Cities identified with high levels of heat vulnerability and exposure may be prioritized for adaptation. The results will contribute to the understanding of city-level heat risks in the developing region of the Asia-Pacific.

Response: We sincerely thank Reviewer #2 for the time and effort in reviewing our manuscript. We really appreciate all the comments and suggestions, as well as the compliments. In our revised manuscript, we carefully took all the comments and suggestions into account. Below is our point-by-point response.

These novel findings will be of interest to others in the region and the wider field. The authors' use of a wide range of data, and the hazard, exposure and vulnerability framework led to a comprehensive evaluation of heat risk. I particularly liked their discussion of urban decongestion and sustainable and regional development. Additionally, besides being expensive, widespread air conditioning usage is particularly energy intensive and exacerbates emissions (if energy is fossil fuel based) leading to positive feedback loops.

Response: Thank you for the positive feedback.

- We really appreciate the comment on widespread air-conditioning, hence, we have decided to include this point in the Discussion section.

Lines 253-261 of the revised manuscript:

More specifically, air-conditioning^{56,58,59} can help them adapt to heat, but their economic capability will be an important factor in this case. It is because the Philippines has among the highest electricity costs in the world. It has the highest electricity costs in Southeast Asia for the commercial and household sectors, even surpassing Singapore⁶⁰. Thus, the Philippine national government, in general, needs to find means and institute measures that could lower electricity costs in the country to help its citizens to be able to adapt to heat increases. That being said, the widespread use of air-conditioning is also energy-intensive and exacerbates emissions (if fossil fuel-based), leading to positive feedback loops. Hence, this should also be considered by the government and the country's energy, manufacturing, commercial and household sectors.

Though there is always room for improvement and the authors have acknowledged some weaknesses in their approach there are a few minor clarifications that would make this work more convincing. The authors could provide some further evidence to strengthen the conclusions.

Thank you for the positive feedback and comments/suggestions. Please see below.

1. The choice of 12 experts and the experts rating of relative weights leads to subjectivity in final HRI ranking. A sensitivity analysis could be helpful in putting these concerns to rest. The authors could show the effects of a slight perturbation (say 5%) in experts' relative weights on the city rankings.

Response: Thank you for the comment/suggestion.

- We have calculated confidence intervals (95% CI) for the resulting risk index values, showing the lower and upper bounds (Table S2).

- More importantly, the risk index has been validated using mortality data (hot dry season) and found a correlation of $r = 0.250-0.239$ for daytime and $r = 0.262-0.314$ for nighttime, both are statistically significant ($p < 0.05$). The possible factors that could have affected the level of correlation have also been discussed. Fig. S1 shows the validation results.

Lines 198-203 of the revised manuscript:

Heat health risk assessment is a multi-disciplinary assessment, and hence cuts across various fields, requiring diversity of expertise. Thus, while the derived weights and the risk assessment results were not consistent among experts, this result is not completely unexpected as it reflects a diversity of views from a set of experts in various, but related, fields. Nevertheless, the overall HHRI was positively correlated with the number of deaths attributable to heat (NDAH), both for daytime ($r = 0.253-0.335$, $p < 0.05$) and nighttime ($r = 0.265-0.321$, $p < 0.05$) (Fig. S1).

Lines 306-312 of the revised manuscript:

Some factors that could have influenced this level of correlation include: (1) the coarseness of the mortality data used (province level, instead of city level), (2) temporal inconsistency between the mortality data (c. 2010) and the HHRI (c. 2015), (3) cloud commination on the remote sensing data, which affected the level of completeness of the datasets used, and (4) the omission of other potentially relevant vulnerability indicators due to data limitation (more on this below). Nevertheless, the validation results are indicative of the accuracy, usefulness and potential of the risk assessment approach employed, as well as the indicators used.

Lines 605-625 of the revised manuscript:

Validation and sensitivity analysis. *Plotting the derived HHRI values of the cities against an observed heat health-related impacts, e.g. recorded heat-related mortality⁵⁶ or summer hospital visits⁴⁴, can provide a direct validation of the risk assessment results. Here, due to the lack of data at the city level for the year c. 2015, we used the available province level mortality data during the hot dry season of 2009-2011. The 32 provinces with mortality data (Fig. S4a) altogether cover 65 cities. To estimate the mortality at the city level, we calculated the ratio of the city's population (2010) to the total population of the province (2010) where the city belongs. This ratio was used to estimate the city's share of mortality from the province's annual average mortality during the period. Using Eq. (3), we estimated the city level number of deaths attributable to heat (NDAH). Finally, scatter plots were produced and correlations between the HHRI and the NDAH were derived (Fig. S1).*

$$NDAH = ((RR - 1)/RR) \times M \quad (3)$$

where $((RR - 1)/RR)$ is the attributable fraction, interpreted as the excess risk due to heat, and M is the city's share of mortality from the province's mortality. The relative risk (RR) was derived based on Eq. (2).

For the sensitivity analysis, we calculated the 95% confidence interval (CI) of the derived HHRI values of all the cities (Table S2). We also performed a sensitivity analysis for the number of lag days in deriving the RR curves and MMTs (Fig. S2). A comparative analysis between the RR in hot dry season and the RR in hot rainy season was also performed (Fig. S6). Finally, the spatiotemporal completeness of the MODIS LST data used was also determined (Figs. S4 and S7).

2. The vulnerability components encompass the domains of age, green space and income (poverty, per capita income). Other studies of heat vulnerability indices typically include some measure of health, social stratification (race / caste), household amenities, education, occupation etc. While some of these variables (such as household amenities) can be proxied by income, the authors should justify and explain the omission of others.

Response: Thank you for the comment and suggestion.

- We completely agree that there could be some other potentially relevant vulnerability indicators that might have been omitted. We explained in the original manuscript, also in the revised manuscript, that the final list of indicators used was influenced by data availability. Nevertheless, we took this comment into account by carefully clarifying this issue in the revised manuscript.

Lines 361-373 of the revised manuscript:

We recognize that our heat risk assessment approach (Fig. 1c; see Methods) has also several limitations, which could be a source of uncertainties in the results. In general, these limitations relate to the unavailability of detailed data, e.g. city-level data on heat-related mortality or hospital visits during the hot dry season (c. 2015). Similarly, detailed data are needed for other potentially relevant indicators for heat vulnerability assessment: sensitivity (e.g. data on pre-existing health conditions^{58,59}) and capacity (e.g. data on proportion of budget allotted to health care services and environmental protection, air-conditioner usage^{56,58,59} and education or literacy rate^{56,58,59}). We are aware that air-conditioner usage and literacy rate in the Philippines are still dependent on the socioeconomic status of each family or individual, unlike in the developed regions. Hence, based on this local knowledge, per capita net income has been used as a proxy indicator for these variables. Nevertheless, all these mentioned indicators, including those deemed relevant but were not mentioned, need to be explored and, where possible, included in the future updates of this study once the needed data at the city level become available.

On a subjective note, this paper will influence thinking in the field. The approach, accounting for limitations, is flexible and can be applied in other countries and settings. Evidence from developing countries, at the forefront of this threat, is less and therefore this paper fulfills that critical gap in knowledge.

Response: Thank you for the positive feedback.

- We really appreciate the positive feedback, and that we have decided to include them in the last part of the Discussion section.

Lines 397-402 of the revised manuscript:

Finally, we recognize that to help advance scientific knowledge and improve understanding in the field of heat health risk, there is still a need for more local level assessments in the developing regions, which are at the forefront of the threat of the combined effects of the UHI phenomenon due to rapid, poorly planned and spontaneous urbanization and heat increases due to climate change. This study aims to contribute toward this direction to help fill and narrow this critical gap in knowledge.

The statistical analysis used is appropriate for this work. The authors have use publicly available data and therefore this work is likely reproduceable by other researchers in the field.

Response: Thank you for the positive feedback.

Again, thank you very much for the time, effort and expertise shared. We sincerely appreciate them.

Reviewers' comments:

Reviewer #1 (Remarks to the Author):

I thank the authors for their careful consideration of all comments. My concern regarding the covered study period and the method of exposure assessment nevertheless remains in the new MS.

First, the authors argued it's correct to use population density rather than total population in the exposure assessment. I could not agree. Simply put, the authors used the city level number of deaths attributable to heat (NDAH) to validate the risk map. But the attributable deaths are directly related to the total population but not the population density. This can be seen in Eq 3. I also think this may be the potential reason of low correlation between NDAH and the risk estimates.

Second, the authors argued that the study period in hazard assessment did not cover the hot rainy season (June-August) mainly due to the cloud effect in LST images and the hotter temperatures in hot dry season compared to hot rainy season. However, in Fig.S5, both daily maximum and minimum temperatures in June (hot rainy season) are higher than them in March (how dry season). The authors need to clearly refine their scientific question and research innovation of this study. If the key innovation is the risk assessment framework and assessment method of disaggregated risk components, it's ok to conduct the hazard assessment in the present way but need to point out the limitation of the covered study period. If the key innovation is to accurately assess the comprehensive heat health risk in Philippine cities, temperature data in March-May during three years is insufficient for such a study.

Reviewer #2 (Remarks to the Author):

The authors have addressed all the comments and suggestions to my satisfaction. I wish them well and look forward to reading more of their work in future!

Gulrez Shah Azhar
Senior Fellow, IHME
University of Washington

Reviewer #1 (Remarks to the Author):

I thank the authors for their careful consideration of all comments. My concern regarding the covered study period and the method of exposure assessment nevertheless remains in the new MS.

Response: Once again, we sincerely thank Reviewer #1 for the time and effort in reviewing our revised manuscript. Like in the previous revision, we carefully took all the comments and suggestions into account in this current version. Below is our point-by-point response.

First, the authors argued it's correct to use population density rather than total population in the exposure assessment. I could not agree. Simply put, the authors used the city level number of deaths attributable to heat (NDAH) to validate the risk map. But the attributable deaths are directly related to the total population but not the population density. This can be seen in Eq 3. I also think this may be the potential reason of low correlation between NDAH and the risk estimates.

Response: Thank you very much for the critical comments. These have all been carefully considered.

(1) On the mismatch or inconsistency of data format in the derivation of exposure layer and validation process – we sincerely thank Reviewer #1 for pointing this out to us. We understand and we also agree. Hence, in this version of the manuscript, we have re-worked the validation part. The NDAH is now expressed into two ways: (i) as a proportion of each city's total population, and (ii) as a density at city level. With this revision, the data format for the validation is now consistent with the data format used to derive the exposure layer. The use of the two indicators (proportion and density) also tests the sensitivity of the validation process.

Lines 613-637 of the revised manuscript.

Validation and sensitivity analysis. Plotting the derived HHRI values of the cities against an observed heat health-related impacts, e.g. recorded heat-related mortality⁵⁶ or summer hospital visits⁴⁴, can provide a direct validation of the risk assessment results. Here, due to the lack of data at the city level for the year c. 2015, we used the available province level all-cause mortality data during the hot dry season of 2009-2011. The 32 provinces with all-cause mortality data (Fig. S4a) altogether cover 65 cities. To estimate the all-cause mortality at the city level, we calculated the ratio of the city's population (2010) to the total population of the province (2010) where the city belongs. This ratio was used to estimate the city's share of all-cause mortality (M) from the province's average all-cause mortality during the period. Using Eqs. (2) and (3), we estimated the city level number of deaths attributable to heat (NDAH), expressed as a proportion of city's total population (%) and as a density at city level (km^{-2}). The use of these two indicators also tests the sensitivity of the validation. The use of density at city level is consistent with the heat exposure index which was produced by using population density (see above). Finally, scatter plots were produced and correlations (Pearson's r) between the HHRI and the NDAH were derived (Fig. S1).

$$NDAH_{\text{city}}(\%) = (x/P) \times 100 \quad (3)$$

$$NDAH_{\text{city}}(\text{km}^{-2}) = x/A \quad (4)$$

where x is the number of deaths attributable to heat at city level expressed as $((RR - 1)/RR) \times M$, where $((RR - 1)/RR)$ is the attributable fraction, interpreted as the excess risk due to heat,

and M is the city's share of all-cause mortality from the province's all-cause mortality. P and A refer to the city's total population and land area, respectively. Here, for x , the average between day and night was used. This was in order for the validation to be consistent with the way the heat hazard index was produced (see above). The relative risk (RR) was derived based on Eq. (2).

The resulting correlation coefficient (r) has increased to 0.359 ($p = 0.003$) for the HHRI vs. proportion of NDAH to city's total population, and 0.436 ($p = 0.000$) for the HHRI vs. density of NDAH at city level. Please see revised Fig. S1 of the current version.

(2) On the use of population density in deriving the exposure layer, instead of total population – we respect the opinion of Reviewer #1 arguing for the use of total population. However, we would like to maintain our view that the use of population density is also methodologically sound for the purpose of this study for the following reasons: (i) an overall composite risk index that could be used for inter-city comparison is derived (HHRI), and for inter-city comparison, population density is the more appropriate indicator to be used than total population; and (ii) various heat health risk studies and other climate-related risk and vulnerability assessments have also used population density instead of total population (e.g. Tomlinson et al. 2011; Buscail et al. 2012; Wolf and McGregor 2013; Morabito et al. 2015; Quader et al. 2017). The recently developed guideline that operationalizes the IPCC risk concept also uses population density (number of farmers per km²) as one of its exposure indicators in its risk assessment example (risk of loss of agricultural livelihoods due to salinity) (GIZ, EURAC & UNU-EHS 2018). Similarly, the World Risk Index also uses normalized population-related indicators, instead of absolute population (e.g. Susceptibility Indicators: share of a population without access to basic sanitation services; share of a population without access to basic drinking water services; and share of a population living below the minimum level of dietary energy consumption) (WRI 2019).

Cited References

- Buscail, C. et al. Mapping heatwave health risk at the community level for public health action. *International Journal of Health Geographics* **11**, 38 (2012).
- GIZ, EURAC & UNU-EHS. *Climate Risk Assessment for Ecosystem-based Adaptation: A guidebook for planners and practitioners*. (GIZ, Bonn, 2018).
- Morabito, M. et al. Urban-hazard risk analysis: Mapping of heat-related risks in the elderly in major Italian cities. *PLoS One* **10**, e0127277 (2015).
- Quader, M.A. Assessing risks from cyclones for human lives and livelihoods in the coastal region of Bangladesh. *International Journal of Environmental Research and Public Health* **14**, 831 (2017).
- Tomlinson, C. J. et al. Including the urban heat island in spatial heat health risk assessment strategies: a case study for Birmingham, UK. *International Journal of Health Geographics* **10**, 42 (2011).
- Wolf, T. & McGregor, G. The development of a heat wave vulnerability index for London, United Kingdom. *Weather Clim. Extrem.* **1**, 59–68 (2013).
- WRI (World Risk Index). Methodological notes of the WorldRiskIndex. (2019). <https://weltrisikobericht.de/english-2/>

In **Fig. 1** below, we illustrate the two cases: use of total population vs. use of population density.

- (i) On the upper row is 'exposure' (Panels A & C) and on the lower row is 'hazard' (Panels B & D).
- (ii) On the left column is with the 'use of total population and total temperature' (Panels A & B) and on the right column is with the 'use of gridded population density and gridded temperature' (Panels C & D).

- (iii) For each Panel (A, B, C & D), there are two cities, a Small city (above) and a Big city (below).
- (iv) In Panel A, the population of the grids are summed to derive the respective total population of the two cities. Should the total population of the two cities together with those of the other cities be later normalized, the Small city would have a lower exposure index than the Big city.
- (v) To be consistent methodologically, i.e. if total population were to be used like in Panel A, the hazard layer in Panel B also needs to be processed in the same way (here, assumed no temperature thresholds). This would result in a consistent lower hazard index for the Small city.

Fig. 1. The use of total population vs. the use of gridded population density.

- (vi) However, in index development for inter-city comparison, it does not make so much sense to think that a big or large city with more people (higher total population) should automatically have higher exposure index, or that a big or large city with higher total temperature due to its larger land area should automatically have higher hazard index.
- (vii) The logic of 'total' would make a big or large city with a much lower surface temperature across its grids due to its urban green spaces to be even higher in terms of heat hazard index because of its larger land area than a small city with a much higher surface temperature across its grids due to high density of impervious surfaces (indicated

by higher population density). This analogy is illustrated by the left column (Panels A & B).

- (vii) In Panel C, the population density values of the grids across the two cities in Panel A are first normalized to a 0-1 index range (0 – lowest exposure index; 1 – highest exposure index) before the average exposure index at city level is derived. This results in a higher exposure index for the Small city.
- (viii) In panel D, the same method is used to process the hazard layer, resulting in a consistent higher hazard index for the Small city.
- (ix) Our approach follows the logic of the right column (Panels C & D), which we believe a more appropriate approach for our derivation of the heat health risk index (HHRI) for the Philippine cities. The main difference between this example and our study is that we used thresholds (MMTs) in our study during the normalization process for the heat hazard layer.

Second, the authors argued that the study period in hazard assessment did not cover the hot rainy season (June-August) mainly due to the cloud effect in LST images and the hotter temperatures in hot dry season compared to hot rainy season. However, in Fig.S5, both daily maximum and minimum temperatures in June (hot rainy season) are higher than them in March (how dry season). The authors need to clearly refine their scientific question and research innovation of this study. If the key innovation is the risk assessment framework and assessment method of disaggregated risk components, it's ok to conduct the hazard assessment in the present way but need to point out the limitation of the covered study period. If the key innovation is to accurately assess the comprehensive heat health risk in Philippine cities, temperature data in March-May during three years is insufficient for such a study.

Response: Thank you very much for the comments. These have all been carefully considered.

- (1) First, we would like to mention that we have already clarified in the manuscript (even in the previous versions) that our study focuses only on the hot dry season. We believe this passage is already comprehensible even to a non-specialist, to mean that the study does not cover the other seasons nor perform an inter-seasonal comparative analysis (see also Fig. 1). Below are some passages from the current version of the manuscript (also present in the previous versions) where “hot dry season” is highlighted.

Lines 5-7 of the current version of the manuscript:

*Here, by applying the risk framework of the Intergovernmental Panel on Climate Change and focusing on a **hot dry season**, we assess the current heat health risk in Philippine cities, whose population accounts for over 40% of the country's total population.*

Lines 76-79 of the current version of the manuscript:

*Thus, drawing upon the concept of risk as a function of hazard, exposure and vulnerability, and the latter as a function of sensitivity and capacity^{28,37}, we assess the current heat health risk in Philippine cities during a **hot dry season** using remote sensing and social-ecological data (Fig. 1; see *Methods*), and discuss the implications of the findings for adaptation planning.*

Lines 526-528 of the current version of the manuscript:

*In this study, we focused on the current (c. 2015) heat health risk during the **hot dry season** in Philippine cities (Fig. 1b).*

- (2) Second, in addition to the two reasons mentioned above by Reviewer #1 (as explained in our manuscript, both in previous and current versions) why this study focuses only on the hot dry season, our analysis also revealed that relative risk to heat is higher during the hot dry season than during the hot rainy season (see Fig. S6). It is unfortunate that this had not been mentioned by Reviewer #1.

Lines 529-531 of the current version of the manuscript:

Furthermore, while there could still be heat health risk during the hot rainy season, our analysis revealed that relative risk to heat is higher during the hot dry season (Fig. S6).

Lines 641-642 of the current version of the manuscript:

A comparative analysis between the RR in hot dry season and the RR in hot rainy season was also performed (Fig. S6).

- (3) Third, we respect the opinion of Reviewer #1 (“...temperature data in March-May during three years is insufficient...”). However, Reviewer #1 should have also realized that, in addition to hot dry season, this study also focuses only on the “current”, i.e. c. 2015. This has been mentioned several times in the manuscript (even in the earlier versions). The inclusion of 2014 and 2016 addresses the issue of the potential natural inter-annual variation in temperature, which was raised during the first round of review. However, we have also explained in the previous response document and in the previous revised version (and also in this current version) that the 2014-2016 period (i.e. not so wide in time coverage) was also considered so that the resulting heat hazard layer from the LST data (2014-2016) would still be temporally consistent with the social-ecological datasets used for the exposure and vulnerability layers (c. 2015). We think that 2013 and/or 2017 and beyond would already be too distant from the target year 2015 considering that this study also uses other social and ecological datasets whose values also vary temporally. Furthermore, such a wide time period may also lead to a multi-temporal risk analysis, which is not the purpose of the study. Below are some passages from the current version of the manuscript (also present in the previous versions) where “current” is highlighted.

Lines 5-7 of the current version of the manuscript:

*Here, by applying the risk framework of the Intergovernmental Panel on Climate Change and focusing on a hot dry season, we assess the **current** heat health risk in Philippine cities, whose population accounts for over 40% of the country’s total population.*

Lines 76-79 of the current version of the manuscript:

*Thus, drawing upon the concept of risk as a function of hazard, exposure and vulnerability, and the latter as a function of sensitivity and capacity^{28,37}, we assess the **current** heat health risk in Philippine cities during a hot dry season using remote sensing and social-ecological data (Fig. 1; see *Methods*), and discuss the implications of the findings for adaptation planning.*

Lines 526-528 of the current version of the manuscript:

*In this study, we focused on the **current (c. 2015)** heat health risk during the hot dry season in Philippine cities (Fig. 1b).*

Still on the viewed insufficiency of the study period (the whole hot dry season over a 3-year period)

– In an earlier heat health risk study in Zhejiang Province, China, Hu et al. (2017) also focused on a study period of three months, while other studies have focused only on one time point (during an extreme heat event) (e.g. Tomlinson et al. 2011; Buscail et al. 2012; Chen et al. 2018). Our consideration of the whole dry season is consistent with Hu et al. (2017) as the whole hot dry season for our case also lasts for three months (March-May). The only difference is that we have a shorter time span of 3 years, compared to 6 years in Hu et al. (2017). Our justifications for this 3-year period have been discussed above (and below). Thus, we believe our study period covering the whole hot dry season over a three-year period is also methodologically sound.

Lines 535-540 of the current version of the manuscript.

In a separate study in Zhejiang Province, China, a three-month study period has also been considered⁵⁶. The only difference between this current study and the said other study is that the latter considered a longer time span of six years⁵⁶. For this current study, the 2014-2016 period was considered so that the resulting heat hazard layer from the LST data would still be temporally consistent with the social-ecological datasets used for the exposure and vulnerability layers (c. 2015) (Fig. 1; Table S1).

Lines 351-354 of the current version of the manuscript.

Daily surface temperature over the whole hot dry season was also used (Fig. 1b), and this is also essential considering that the use of a single time-point may not be able to capture and represent the heat health risk over a certain period or season, but rather limited only to a particular point in time, such as during an extreme heat event⁴⁰⁻⁴².

- (4) Fourth, empirically, this study is original. For its innovation in terms of methods, please see Lines 318-354 of the current version of the manuscript (these have also been discussed in the previous revised version).
- (5) Fifth, for the extensive discussion of the limitations of the study, please see Lines 306-312 and Lines 365-388 of the current version of the manuscript (these have also been discussed in the previous revised version). We believe that the limitation on time period is no longer necessary because it is already mentioned clearly and outright (in the beginning) that the study focuses only on the “current” and “hot dry season”.

Again, we sincerely appreciate the time, effort and expertise shared.

Cited References

- Buscail, C., Upegui, E. & Viel, J.-F. Mapping heatwave health risk at the community level for public health action. *Int. J. Health Geogr.* **11**, 38 (2012).
- Chen, Q., Ding, M., Yang, X., Hu, K. & Qi, J. Spatially explicit assessment of heat health risk by using multi-sensor remote sensing images and socioeconomic data in Yangtze River Delta, China. *Int. J. Health Geogr.* **17**, 15
- Hu, K., Yang, X., Zhong, J., Fei, F. & Qi, J. Spatially explicit mapping of heat health risk utilizing environmental and socioeconomic data. *Environ. Sci. Technol.* **51**, 1498–1507 (2017).
- Tomlinson, C. J., Chapman, L., Thornes, J. E. & Baker, C. J. Including the urban heat island in spatial heat health risk assessment strategies: a case study for Birmingham, UK. *Int. J. Health Geogr.* **10**, 42 (2011).

Reviewer #2 (Remarks to the Author):

The authors have addressed all the comments and suggestions to my satisfaction. I wish them well and look forward to reading more of their work in future!

Gulrez Shah Azhar
Senior Fellow, IHME
University of Washington

Response: Again, we sincerely appreciate the time, effort and expertise shared.

REVIEWERS' COMMENTS:

Reviewer #1 (Remarks to the Author):

Thanks to the authors for their thorough revision of the manuscript (MS). This is a commendable work after two major revisions, and the analyses were mostly done properly.

My only concern is about the validation part of using NDAH_city (%). The author should remove this index, because NDAH_city (%) is logically related to the combination of hazard and vulnerability rather than exposure. Inter-city comparison of NDAH_city (%) is similar to the comparison of attributable mortality fractions to heat, which is a function of heat hazard and heat vulnerability. The current heat risk should be validated by NDAH_city (km²) correctly.

Point-by-point Response to Review Comments/Suggestions
NCOMMS-19-33548B

Reviewer #1 (Remarks to the Author):

Thanks to the authors for their thorough revision of the manuscript (MS). This is a commendable work after two major revisions, and the analyses were mostly done properly.

Response: Once again, we sincerely thank Reviewer #1 for the time and effort in reviewing our revised manuscript. Like in the previous revision, we carefully took all the comments and suggestions into account in this current version. Below is our point-by-point response.

My only concern is about the validation part of using NDAH_city (%). The author should remove this index, because NDAH_city (%) is logically related to the combination of hazard and vulnerability rather than exposure. Inter-city comparison of NDAH_city (%) is similar to the comparison of attributable mortality fractions to heat, which is a function of heat hazard and heat vulnerability. The current heat risk should be validated by NDAH_city (km²) correctly.

Response: Thank you for the suggestion. It has been followed.

Again, we sincerely appreciate the time, effort and expertise shared.